

# Forward Modelling of SAR Backscatter during Lake Ice Melt Conditions using the Snow Microwave Radiative Transfer (SMRT) Model

Justin Murfitt[1,2*], Claude Duguay[1,2], Ghislain Picard[3], Juha Lemmetyinen[4]

[1]Department of Geography and Environmental Management, University of Waterloo, 200 University Ave W, Waterloo, Canada, N2L 3G1
[2]H2O Geomatics Inc., Waterloo, ON N2L 1S7, Canada
[3]Institute of Environmental Geosciences, Université Grenoble Alpes, 38402 Grenoble, France
[4]Finnish Meteorological Institute, Helsinki, Finland

*Correspondence to*: Justin Murfitt (jmurfitt@uwaterloo.ca)

**Abstract.** Monitoring of lake ice is important to maintain transportation routes but in recent decades the number of in situ observations have declined. Remote sensing has worked to fill this gap in observations, with active microwave, particularly synthetic aperture radar (SAR), being a crucial technology. However, the impact of wet conditions on radar and how interactions change under these conditions has been largely ignored. It is important to understand these interactions as warming conditions are likely to lead to an increase in the occurrence of slush layers. This study works to address this gap using the

snow microwave radiative transfer (SMRT) model to conduct forward modelling experiments of backscatter for Lake Oulujärvi in Finland. Experiments were conducted under dry conditions, under moderate wet conditions, and under saturated conditions. These experiments reflected field observations during the 2020-2021 ice season. Results of the dry snow experiments support the dominance of surface scattering from the ice-water interface. However, conditions where layers of wet snow are introduced show that the primary scattering interface changes depending on the location of the wet layer. The

addition of a saturated layer at the ice surface results in the highest backscatter values due to the larger dielectric contrast created between the overlying dry snow and the slush layer. Improving the representation of these conditions in SMRT can also aid in more accurate retrievals of lake ice properties such as roughness, which is key for inversion modelling of other properties such as ice thickness.

## 1 Introduction

There is increasing interest in the study of lake ice across different latitudes. Both lakes and lake ice act as important controls of local climate and energy balance, impacting local precipitation amounts and temperatures (Rouse et al., 2008; Baijnath-Rodino et al., 2018; Eerola et al., 2014). Beyond the impact on local conditions, lake ice provides crucial ecological services across the northern hemisphere. The formation of lake ice is crucial to the establishment of ice roads which aid in the transportation of goods and people during winter months. The majority of the Tibbitt to Contwoyto Winter Road in the





Northwest Territories, Canada is constructed over ice and provides a supply line to mining operations; over 3800 tonnes of material was hauled on the road during the 2020 season (2020 Winter Road Poster, 2020). The formation of lake ice at mid-latitudes is also important for recreational activities such as ice fishing and snowmobiling which are major contributors to local economies in winter months (Cummings et al., 2019). Furthermore, under the World Meteorological Organizations Global Climate Observation System (GCOS) both lake ice cover and lake ice thickness are named as thematic products of lakes as an

essential climate variable (ECV), meaning that it is important to have accurate and consistent records of changes in these variables to act as indicators of climate change (World Meteorological Organization, 2022).

While long-term ice phenology records >500 years do exist for a small sample of lakes (Sharma et al., 2022), observations of lake and river ice phenology have been declining since the 1980s (Murfitt and Duguay, 2021). These observations remain critical as the warming climate is resulting in later ice formation and earlier ice decay, leading to shorter ice seasons (Hewitt,

2019; Lopez et al., 2019). These patterns are expected to continue into the future under different climate modelling scenarios (Brown and Duguay, 2011; Dibike et al., 2012). One possible issue for the use of lake ice with increasing temperatures is that the occurrence of mid-winter melt events and slushing events, most common at mid-latitudes (Ariano and Brown, 2019), will increase. Increased occurrence of these events could pose serious safety risks to the use of lake ice for transportation and recreation.

Although ground observations of lake ice phenology have declined in recent decades, the use of remote sensing technologies has become more popular. Optical, passive microwave, and active microwave have shown to be capable of determining lake ice phenology dates and ice cover extents (Wu et al., 2021; Du et al., 2017; Hoekstra et al., 2020). Active microwave data, specifically synthetic aperture radar (SAR), is the most popular choice of the three due to several reasons. For example, unlike optical imagery, it does not require sunlight to image the ice surface and can do so under most weather conditions. Additionally,

SAR imagery provides resolutions of <50 m for most image products allowing for the delineation of small and medium lakes (Murfitt and Duguay, 2021).  In recent years there has been a shift in the understanding of how active microwave signals interact with lake ice. Initial investigations of lake ice in the 1980s using X and L-band side-looking airborne radar systems connected high radar returns to the presence of tubular bubbles in the ice, stating that bright signals in the imagery were due to a double-bounce scattering mechanism (Weeks et al., 1981). This double-bounce was created as the radar signal interacted

with the vertical tubular bubbles and then with the ice-water interface where there is a high dielectric contrast between the ice and water. Further investigations using spaceborne C-band systems (ERS-1 and RADARSAT-1) continued to support this theory and quantified the backscatter observed from lake ice (Jeffries et al., 1994; Duguay et al., 2002). However, in more recent years with the advent of fully polarimetric SAR data, new research has analyzed the scattering contributions from lake ice and determined that the dominant mechanism is a single bounce or surface scattering mechanism (Atwood et al., 2015;

Engram et al., 2012; Gunn et al., 2018). This is attributed to roughness at the ice-water interface. Explanations for the roughness at this interface include the presence of tubular bubbles in the lower layers of the ice, methane ebullition bubbles, and differing rates of ice growth (Gunn et al., 2018; Engram et al., 2012, 2013).



Modelling approaches provide a further valuable opportunity to explore the impact of changing ice properties (ice thickness, roughness, bubble size) on backscatter from lake ice. The results of recent modelling studies support the new polarimetric

decomposition results, finding that the presence of elongated tubular bubbles at the ice-water interface has little impact on the backscatter from lake ice and that roughness of the ice-water interface is the key factor (Atwood et al., 2015; Tian et al., 2015). However, past modelling approaches have several limitations where applied models ignore different aspects of the lake ice column. For example, models have not factored in the presence of snow, ignored roughness of different interfaces, or considered the ice column to be a single homogenous layer. Recently the snow microwave radiative transfer (SMRT) model

was used to conduct sensitivity analysis for lake ice under dry conditions (Murfitt et al., 2022, 2023). SMRT provides a framework where different electromagnetic, microstructure, and interface modules can be used, allowing for more faithful modelling of real conditions. Results of a recent sensitivity analysis using SMRT were found to be consistent with the results of other lake ice modelling and satellite observations, supporting the crucial role of the ice-water interface (Murfitt et al., 2022, 2023).

While recent modelling has focused on the roughness of the ice-water interface, increased water content and the representation of melt conditions has been largely ignored. Wakabayashi et al. (1999) used the integral equation model (IEM) for surface scattering to investigate scattering mechanisms from floating and grounded ice cover on the North Slope of Alaska. Simulations indicated that for both ice conditions the inclusion of water on the ice surface, either in a limited area or across the entire ice cover, results in an increase in backscatter (Wakabayashi et al., 1999). More recently, Han and Lee (2013) investigated the

role that ice phase transition plays on backscatter due to changes that occur in the dielectric constant and roughness conditions using a ground-based C-band scatterometer. Similar to Wakabayashi et al. (1999), IEM was also used for the experiments conducted on Chuncheon Lake in South Korea. Changes of phase were simulated by spreading a thin layer of water on the top of the ice cover and analysis showed that when the water was initially spread on the ice, scattering from the top of the ice surface was strong due to the higher dielectric constant between water and air than ice and air (Han and Lee, 2013). However,

as the water froze, ice-bottom and volume scattering increased due to more transmission of the signal through the ice column (Han and Lee, 2013). The experiments in Alaska and South Korea provide important insights into how changes in water content can impact the backscatter signal from lake ice. However, these experiments have certain limitations, for example representations of the snow cover. Further research is still needed to explore a range of ice cover conditions throughout the winter season. Additional sensitivity tests parameterized using collected field data can provide further insights and

confirmation of how radar backscatter over ice cover is impacted by wet conditions.

SMRT allows us to address these gaps and work toward modelling a complete picture of ice conditions during melt events. Previously mentioned lake ice studies using SMRT only evaluated changes in backscatter under dry conditions (Murfitt et al., 2022, 2023). Therefore, experiments during wet conditions are an important next step in the application of this model for lake ice cover. The objective of this paper is to exploit field data collected at Lake Oulujärvi in Finland to determine how changes

in the water content of snow overlying ice cover and the appearance of slush layers impact backscatter. To meet this objective, sensitivity tests are conducted to understand how backscatter changes with increases in snow water content and interface



roughness. These modelling results are compared to observed Sentinel-1 SAR backscatter during different field conditions, providing support for the results of the sensitivity analysis and insight into the connection between ice conditions and backscatter during the 2020-2021 field season.

## 2 Methods

### 2.1 Snow Microwave Radiative Transfer (SMRT) Model

SMRT is an active-passive model for conducting simulations of microwave intensities from snowpacks. A full description of the model can be found in Picard et al. (2018). The model also allows for the inclusion of freshwater and saline ice layers that can be combined with snowpacks to properly represent observed conditions (Soriot et al., 2022; Murfitt et al., 2022, 2023). The model is run within a python environment and allows for user flexibility by allowing model runs to be set using different electromagnetic models (e.g., improved Born approximation or IBA, dense media radiative transfer or DMRT) and microstructure models (e.g., exponential, and sticky hard spheres). The user-selected electromagnetic model is parameterized using the selected microstructure model and user defined properties (medium temperature, thickness, density, volumetric liquid water content, etc.). The electromagnetic model is used to determine the necessary electromagnetic quantities, such as the scattering coefficient, absorption coefficient, and phase matrix (Picard et al., 2018). Roughness and associated reflectivity and transmissivity coefficients within the defined snow and ice columns are set using either Fresnel equations, the integral equation model (IEM), or Geometrical Optics (Picard et al., 2018; Fung et al., 1992; Tsang and Kong, 2001). Roughness can be set between different layers of a single medium, at the interface between two mediums (e.g., snow-ice interface), or between a medium and the underlying substrate (e.g., ice-water interface). SMRT uses the discrete ordinate and eigenvalue (DORT) method to solve the radiative transfer equation once the necessary parameters have been solved in the other components of the model. The user can create a custom sensor to parameterize the model (e.g., specific frequency, incidence angle, and polarization) or choose from a list of pre-defined sensors. Resulting intensity from SMRT can be obtained from all or specific directions from the defined snowpack or ice column (Picard et al., 2018). This study only evaluated modelled co-pol (HH and VV) backscatter because cross-pol (HV and VH) is under modelled with the current implementation of IEM in SMRT which is used for parameterization of interface roughness as discussed below. Additionally, experiments assume that there were no tubular bubbles present in the ice column because SMRT does not currently allow for the inclusion of vertically oriented bubbles and the impact of these bubbles has been demonstrated to be limited (Atwood et al., 2015; Murfitt et al., 2022).

This study uses the IBA for the electromagnetic model in SMRT. This model is utilized to allow for a broader exploration of volumetric water content (VWC) within SMRT. Under dry conditions, the real and imaginary components of permittivity for snow grains and ice mediums is determined using the formulations given in Mätzler et al. (2006). However, this cannot be used for wet snow mediums that mix air, ice, and water. SMRT was recently updated to include several models for addressing the mixing of these different components in snow mediums (Picard et al., 2022b). While the results of these models are different, there is agreement that increases in the VWC increase both the real and imaginary components of a



medium's permittivity. This study uses the MEMLS v3 permittivity model for wet snow conditions (Mätzler and Wiesmann, 2007). Following Picard et al. (2022a), this model was selected because it is based on real measurements and provides reliable performance at higher water content values. Both the sticky hard spheres (SHS) and exponential microstructure models were utilized in this study (Picard et al., 2018, 2022a). SHS was used specifically for ice mediums due to its use in past radiative transfer modeling studies (Gunn et al., 2015; Murfitt et al., 2022). SHS is parameterized using the stickiness parameter ($\tau$) which is a representation of the tendency of spheres within the medium to cluster. Increased values of $\tau$ indicate a lower tendency of spheres to stick together. The microstructure of snow mediums was parameterized using the exponential microstructure model. This model was used due to the availability of detailed snow pit measurements during the 2020-2021 ice season. Density and SSA measurements were used to calculate the effective correlation length ($\boldsymbol{p_{ex}}$) and parameterize the microstructure model, this is discussed further in section 2.5.

Roughness of different interfaces within the ice and snow mediums for this study were represented using IEM with an exponential autocorrelation function. IEM was selected because it is better suited to small roughness values and has been used in previous freshwater ice and snow modelling studies (Gherboudj et al., 2010, Murfitt et al., 2022). As such, this study will focus on variations in small scale roughness of interfaces within the snow and ice column. IEM is parameterized using root mean square height (RMSH) which is the approximation of vertical variation of surface roughness and interface correlation length which is the horizontal displacement between two points on the rough surface (Ulaby and Long, 2014). The validity of IEM is maintained when $k*RMSH < 2$ and $\boldsymbol{k^2 * RMSH * Correlation\ Length} < \sqrt{eps_r}$, where $k$ is the wavenumber and $\boldsymbol{eps_r}$ is the ratio between the permittivity of the mediums at the interface (Fung et al., 1992; Fung and Chen, 2010). The range of IEM is extended in SMRT using the method stated in Brogioni et al. (2010), where the Fresnel coefficients are determined using either the incidence angle ($\boldsymbol{k^2 * RMSH * Correlation\ Length} < \sqrt{eps_r}$) or an angle of $0°$ ($\boldsymbol{k^2 * RMSH * Correlation\ Length} > \sqrt{eps_r}$). This extension was first developed for Advanced IEM (AIEM), as such caution should be used when applying this extension as the precise validity on the original IEM has not been verified.

## 2.2 Study Site

This study focuses on Lake Oulujärvi (27.25° E, 64.29° N), which is one of the largest lakes in Finland (**Figure 1**). Lake Oulujärvi is located 473 km north of Helsinki and the outflow of the lake is to the Oulujoki River which drains into the Gulf of Bothnia (Hyvaerinen, 2004). The lake has a surface area of 928 km$^2$ with a mean depth of 7.6 m that can range up to 36 m (Hyvaerinen, 2004). Historical data from 1854 to 2002 for the lake indicates that the average date of freeze-up ranges from November 10 to 20 and average date of break-up ranges from May 15 to 20 (Korhonen, 2006, 2005). The average thickness of the ice cover (1961 – 2000) on the lake is between 60 and 70 cm (Korhonen, 2006, 2005). However, more recent observations of ice thickness in 2021 report maximums of 42 cm (Weyhenmeyer et al., 2022). Additionally, trends in lake ice cover indicate that the date of ice-on shifted later by 0.8 days/decade and ice-off shifter earlier by 0.9 days/decade between 1853 and 2018 (Sharma et al., 2021). Climate normals (1981 – 2010) for the Kajaani meteorological station, 23 km east of the main basin of





Lake Oulujärvi, show that maximum average temperatures of 20.8 °C occur in July and minimum average temperatures of -15.2 °C occur in January (Pirinen et al., 2012). Maximum snow depth is reported during March, reaching an average of 53 cm during the 1981 – 2010 normals period (Pirinen et al., 2012).

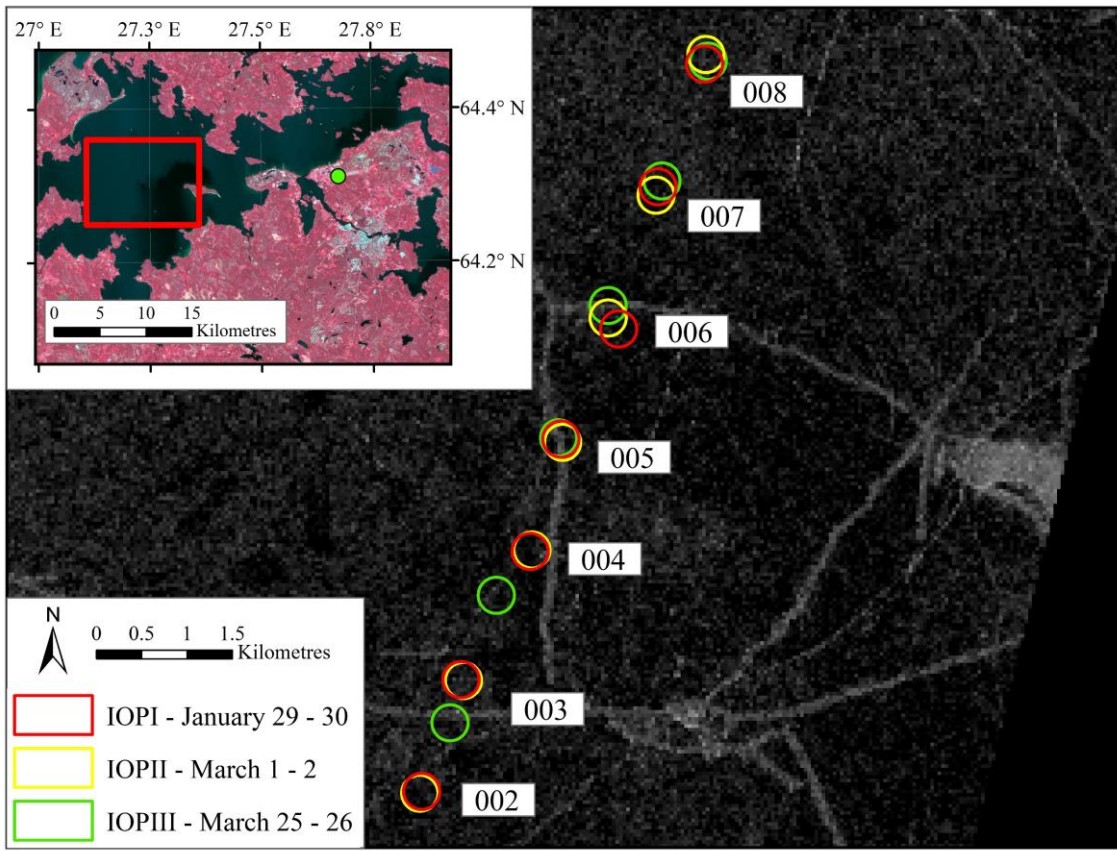

**Figure 1.** In Situ data collection locations on Lake Oulujärvi during the 2020-2021 ice season. The Sentinel-1 image shown was acquired January 29 aligning with IOPI. The area of high backscatter on the east side of the image is an island. Linear features with high backscatter are ice ridges or deformations in the surface. This figure contains Copernicus Sentinel-1 data (2021), processed by ESA. Landsat-8 Level-2 image courtesy of the U.S. Geological Survey.

**2.3 SAR Imagery**

To provide a record of backscatter for the 2020 - 2021 ice season on Lake Oulujärvi and a point of comparison for the modelled results, 151 Sentinel-1 (C-band, 5.405 GHz), 82 Extra Wide (EW) swath HH-pol and 69 Interferometric Wide (IW) swath VV-pol, SAR images were downloaded from the Alaska Satellite Facility (https://asf.alaska.edu/) for the period December 13, 2020, and May 7, 2021. Sentinel-1 EW images have a pixel spacing of 40 m and there was an average of 2 days between images. Sentinel-1 IW, pixel spacing 10 m, were also acquired with a gap of 2.4 days (some days had two images available). The combination of both image sets results in a temporal coverage of 1.09 days for Lake Oulujärvi. The Sentinel-1 images



were preprocessed using the Sentinel Application Platform (SNAP, European Space Agency, 2020), with images being σ° calibrated and speckle filtered with the Refined Lee filter (Lee, 1981). Terrain correction was performed using the ACE30 DEM and both IW and EW images were resampled to a pixel spacing of 40 m to ensure consistency between datasets.

Incidence angles for both datasets ranged from 24 to 43°. This difference in incidence angle resulted in fluctuations in the backscatter extracted from the lake ice sites. To address these fluctuations, linear regression was used to normalize the backscatter. This method has been applied previously for both Arctic sea ice, high Arctic lakes, and mid-latitude small/medium-sized lakes (Murfitt et al., 2018; Murfitt and Duguay, 2020; Mahmud et al., 2016). Two hundred points were randomly generated across Lake Oulujärvi to serve as virtual sample sites. These sample sites were located at least 300 m from shore and not within 40 m of each other to prevent shoreline contamination and repetition of samples. The backscatter and

projected incidence angle were extracted from these sites for each image in the acquired set, resulting in a total sample size of 26,800. Only images where there was full ice cover on the lake were used. The resulting samples were then split into a training and testing set of 70% and 30%, respectively. The training set was then used to develop a linear regression representing the impact of incidence angle on backscatter for Lake Oulujärvi. The linear regression showed a slope of -0.34 dB/° ($R^2$ = 0.26, p < 0.01. RMSE = 3.33 dB). This slope is similar to past corrections for Sentinel-1 imagery acquired over Lake Hazen of -0.35

dB/° (Murfitt and Duguay, 2020). For the correction of IW and EW images, the median incidence angle of all samples was used, which corresponds to 36.31°. Around each of the sites shown in **Figure 1**, a 200 m buffer was generated to extract average, maximum, and minimum normalized linear intensity values that were converted to backscatter (dB).

    **Figure 2** shows an average trend in backscatter for all locations. Additionally, 2 m air temperature extracted from ERA5 is displayed. This specific field sites will be further explored in the discussion section.

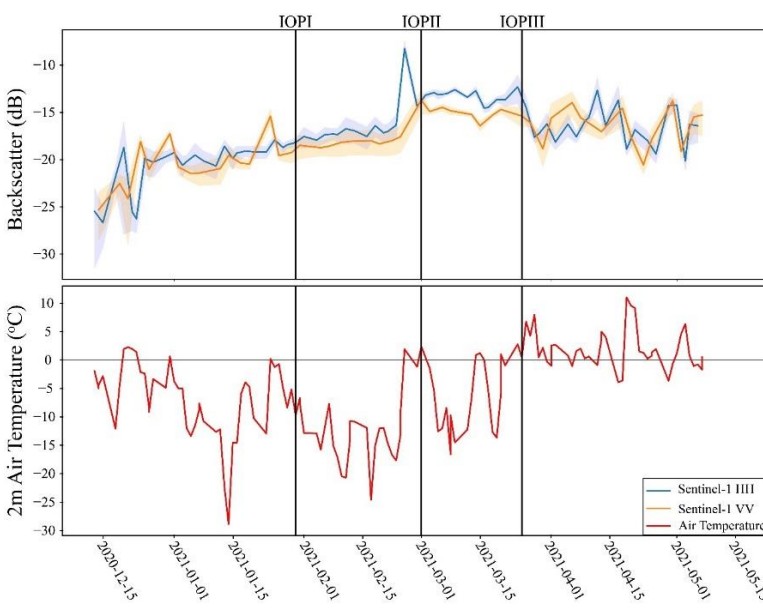


**Figure 2.** Averaged backscatter trends for all sites (top). The shaded area indicates the standard deviation in extracted values. ERA5 2m air temperature is shown on the bottom.





## 2.4 In Situ Data: 2021 Field Campaign

During the 2020 – 2021 ice season, field campaigns were conducted to collect in situ data on snow and ice cover at Lake
Oulujärvi. These field campaigns took place during three observation periods (IOPI, IOPII, and IOPIII). The first observation
period, IOPI, occurred on January 29 and January 30, the second observation period, IOPII, occurred on March 1 and 2, and
the final observation period, IOPIII, occurred on March 25 and 26. For IOPI and IOPIII, measurements of ice thickness and
the thickness of layers (snow and clear ice) were taken approximately every 400 to 900 m. For IOPII, ice thickness
measurements were made every 1200 to 1600 m. In addition to ice thickness measurements, snow depth and bulk density were
also recorded at each of these locations. During IOPII, a SnowHydro Magnaprobe (Sturm and Holmgren, 1999) was used to
collect snow depth measurements approximately every 5 m. For IOPIII, 10 snow depth measurements were recorded using a
manual probe every 100 m along the study transect. For the purposes of this study, only the points shown in **Figure 1** are used.
This is due to the availability of matching snow pit data discussed below. Additionally, the points selected incorporated
locations where both dry snow and wet snow conditions were observed throughout the ice season. A summary of the ice
thickness and snow depth and bulk density measurements collected during each IOP is provided in **Table 1**.

**Table 1.** Summary of snow and ice measurements collected during the 2021 field campaigns on Lake Oulujärvi. Density values show bulk density measurements made at the ice thickness measurement sites.

|  | Ice Thickness (m) | Snow Ice Thickness (m) | Snow Depth (m) | Snow Density (kg m⁻³) |
|---|---|---|---|---|
| IOPI | $0.32 \pm 0.04$ | $0.03 \pm 0.04$ | $0.27 \pm 0.05$ | $185 \pm 34$ |
| IOPII | $0.35 \pm 0.07$ | $0.02 \pm 0.03$ | $0.36 \pm 0.07$ | $245 \pm 28$ |
| IOPIII | $0.42 \pm 0.06$ | $0.06 \pm 0.06$ | $0.31 \pm 0.11$ | $319 \pm 69$ |

In addition to these measurements, detailed snow pit data were collected for the sites identified in **Figure 1**. Data were collected
for each layer of the snowpack through manual means and specific surface area (SSA) was collected using the IceCube
instrument (Gallet et al., 2009). This study relied on the layer properties such as SSA, layer temperature, water content, density,
thickness, maximum and minimum grain size, and grain type. A summary of these data is provided below and differs from the
bulk properties presented in **Table 1**. It should also be noted that the densities from the snow pits are higher than those measured
for bulk density. Additionally, these data were inconsistent and missing for some sites due to conditions not allowing for the
accurate collection of these properties. The collected data were used to investigate the relation between SSA and density as
discussed further in section 2.5. Additionally, these data provided key information about the temperature of the snowpacks.
During IOPI (January 29/30), the average temperature was $270.22 \pm 2.1$ K. The average density was $214.09 \pm 77.6$ kg m⁻³ and
average SSA was $34.34 \pm 11.9$ m² kg⁻¹. For IOPII (March 1/2), temperatures increased to an average of $272.09 \pm 1.5$ K,
temperatures were higher on March 1 reaching 273.2 K. The average snow density for IOPII was $296.9 \pm 47.8$ kg m⁻³, with a
higher average density of 305.5 kg m⁻³ measured March 1 compared to March 2. SSA decreased for IOPII to an average of


14.96 ± 11.9 m² kg⁻¹, average SSA was higher on March 1 at 17.4 m² kg⁻¹. Finally, on IOPIII (March 25/26), temperatures further increased to 272.51 ± 1.0 K. Average density was highest for IOPIII with an average of 335.16 ± 61.5 kg m⁻³. Average SSA was similar to IOPII at 14.46 ± 2.3 m² kg⁻¹, however, was only available for one site.

## 2.5 SMRT Experiments

To study how wet conditions impact the backscatter from lake ice, three distinct numerical experiments were conducted. Each experiment reflects conditions that were observed during these field campaigns in 2021. The field data described in section 2.4 were used to parameterize SMRT for the respective dates when each condition was observed. Additionally, snowpack observations across the lake were used to create a generalized two-layer snowpack for each of the different experiments. For each IOP, the snow depth was determined by averaging the most detailed data available (i.e., manual or Magnaprobe measurements). The temperature of the snowpack was determined by averaging the available measurements from the snow pit analysis. While water content was one of the measured parameters, the values were not successfully recorded and therefore qualitative observations were relied on for determining the state of the snow layers. The key inputs for SMRT are the layer thickness (i.e., snow depth and ice thickness), snow density, $p_{ex}$, layer temperature (i.e., snow and ice temperature), ice porosity, spherical bubble radius, VWC, roughness parameters, and ice layer stickiness. For ice layers, to properly parametrize the sticky hard spheres microstructure model, a value of stickiness (τ) had to be decided. Due to clear ice being considered as inclusion free, the layer was assigned a τ of 1, which corresponds to relatively random positions of the bubbles. Snow ice on the other hand can have an increased presence of spherical bubbles and therefore a value of 0.4 was used. Parameterization of the snow microstructure model will be discussed later in this section.

The majority of parameters were collected during the field campaign, except for ice temperature and $p_{ex}$. Ice temperature was determined using the Canadian Lake Ice Model (CLIMo, Duguay et al., 2003). CLIMo is a 1-D model that uses an unsteady heat conduction equation (Maykut and Untersteiner, 1971) and surface energy budget to determine layer boundary temperatures, ice thickness, and snow depth throughout an ice season. For the purposes of this study, the main output used is the ice layer boundary temperatures. For Lake Oulujärvi, CLIMo is parameterized using both daily averaged ERA5 data (Hersbach et al., 2020) for temperature (C), humidity (%), cloud cover (0-1), and wind speed (m/s). The snow accumulation input for CLIMo was determined using local meteorological data from the Kajaani weather station located 23 km southwest from the field sites. CLIMo was run using a mixing depth of 7 m and a snow density of 185 kg/m³, which reflected values observed for dry snow cover during the 2021 field campaign. Validation data is limited to only dates where field data was collected and showed good comparison with ice thicknesses estimated by CLIMo, with an average difference of 0.01 m. Snow depth in CLIMo is underestimated compared to field measurements, likely due to the distance between the meteorological measurements and the location of the field measurements and a single value for snow density being used, the average difference was 0.13 m. While this difference is large, thermodynamic modelling is the best alternative for determining layer temperatures and was used as the input for ice temperature in SMRT. CLIMo was run to produce a five-layer ice column, however, only





snow and clear ice measurements were taken from Lake Oulujärvi. Therefore, the temperatures for the boundary of layers 1 and 2, counted from the top, were averaged to determine a snow ice layer temperature and the temperatures for the boundaries between layers 2 - 5 were averaged to produce a temperature for the clear ice layer.

To properly parameterize the exponential microstructure model used in SMRT for the snow layers, correlation length, $p_{ex}$, is
needed. Correlation length can be determined using relations determined between this parameter and grain size in Mätzler (2002). Following the SMRT simulations in Rutter et al. (2019), $p_{ex}$ was calculated using **Equation 1**:

$$p_{ex} = 0.75 \left( \frac{4*\left(1-\frac{\rho}{\rho_{ice}}\right)}{SSA*\rho_{ice}} \right) \quad (1)$$

where $\rho_{ice}$ is the density of the ice, assumed to be 916.7 kg m$^{-3}$, $\rho$ is the density of the snow layer, and SSA is the specific surface area (m$^2$ kg$^{-1}$). The empirical coefficient 0.75 was confirmed as a possible value of the microwave polydispersity of
fine grained snow by recent theoretical investigations (Picard et al., 2022a). When SSA values were unavailable, snow density is used to estimate the SSA value through a logarithmic regression. In past SMRT experiments, regressions developed for terrestrial snow have been applied to lake ice snow covers (Murfitt et al., 2022). However, these representations may not reflect the differences between these two systems (i.e., wind redistribution, and differences in snow structure). Therefore, for the SMRT experiments over Lake Oulujärvi, SSA values acquired during these 2021 field campaigns were analysed to develop a
similar regression that is more optimal for Lake Oulujärvi. In total 78 samples were used, and the resulting regression is shown in **Equation 2** ($R^2 = 0.41$), and a scatterplot can be found in **Appendix A**:

$$SSA = -24.00 \ln(\rho) + 156.09 \quad (2)$$

Vargel et al. (2020) developed an equation using observations for terrestrial Canadian Subarctic and Arctic snow, shown by **Equation 3**:

$$SSA = -17.65 \ln(\rho) + 118.07 \quad (3)$$

The difference in coefficients between these two relations indicates that these snow systems may be dissimilar. **Equation 2** was used to determine an SSA values for the generalized snowpack and **Equation 1** was used to calculate the $p_{ex}$ for each of the experiments. It should be noted that under dry snow conditions, there is likely very little impact of differing SSA values on C-band backscatter.


For use in SMRT the generalized snowpack was split into two layers. Two layers were used so that wet snow could be added at different depths in the snowpack (section 2.5.2). The following subsections detail the experiments performed during the different observation periods for the 2020-2021 ice season. **Figure 3** below provides a graphical representation of these different experiments indicating the location of wet snow layers and rough interfaces using Site 005 (see **Figure 1**) as an
example. **Table 2** details the snow and ice layer parameters for the different IOPs and **Table 3** shows the ice thicknesses for each site across the IOPs.





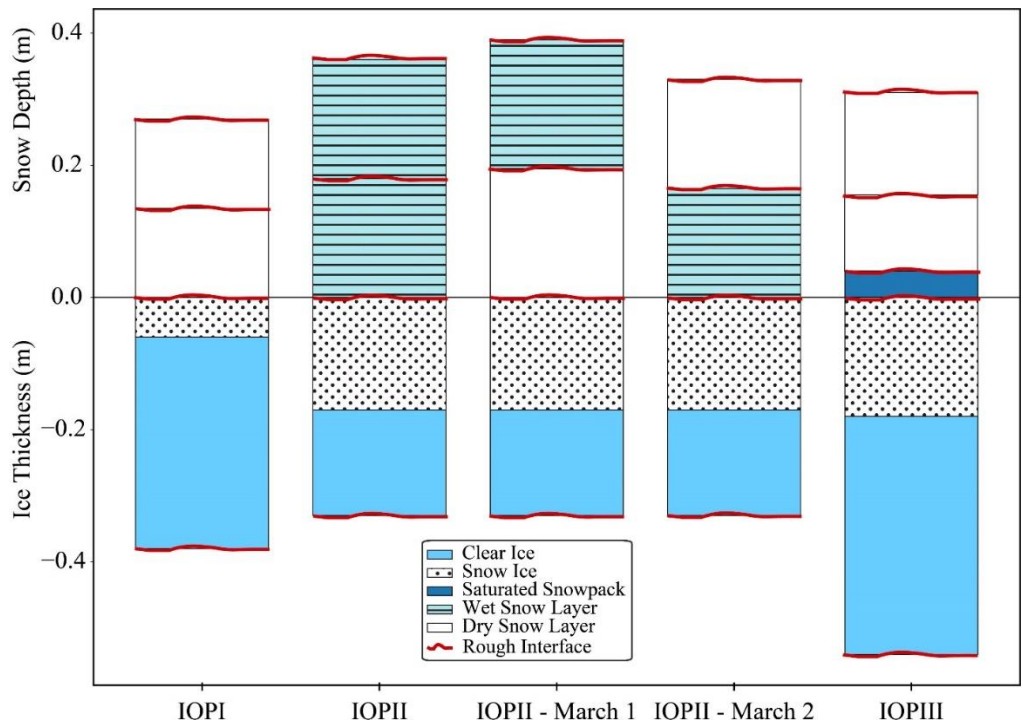

**Figure 3.** SMRT simulations for the different experimental conditions using observations from these 2021 field campaigns. The ice columns displayed are representative of Site 005 (Figure 1). Red lines indicate where rough interfaces were added for each of the different scenarios.

**Table 2.** Constant snow and ice parameters for during IOPs conducted for the 2021 ice season.

| Physical Property | IOPI | IOPII | IOPIIa | IOPIIb | IOPIII |
|---|---|---|---|---|---|
| Snow Depth (m) | 0.27 | 0.36 | 0.39 | 0.33 | 0.31 |
| Wet Snow Density (kg m$^{-3}$) | | 297 | 321 | 218 | 355 |
| Dry Snow Density (kg m$^{-3}$) | 185 | | 270 | 296 | 317 |
| Wet Snow SSA (m$^2$ kg$^{-1}$) | | 15 | 18 | 10 | 16 |
| Dry Snow SSA (m$^2$ kg$^{-1}$) | 30 | | 15.47 | 13.95 | 18.34 |
| Wet Snow Temperature (K) | | 273.2 | 273.2 | 273.2 | 273.2 |
| Dry Snow Temperature (K) | 270.2 | | 273 | 271 | 271.9 |
| Snow Ice Temperature (K) | 272.9 | 272.8 | 272.8 | 272.8 | 272.8 |
| Clear Ice Temperature (K) | 272.9 | 272.9 | 272.9 | 272.9 | 273 |
| | | | | | |
| Constant Snow RMSH (mm) | | | 1.00 | | |
| Constant Snow Correlation Length (mm) | | | 10.00 | | |





| Snow-Ice Correlation Length (mm) | | 50.00 |
| Correlation Length Ice-Water Interface (mm) | | 70.00 |
| Bubble Radius (mm) | | 1.00 |
| Snow Ice Porosity (%) | | 10 |

**Table 3.** Snow and clear ice thickness observed during each IOP. The flooded snow column indicates which sites in IOPIII had slush formed at the snow-ice interface.

| Ice Thickness (m) | IOPI | | IOPII | | IOPIII | | |
| --- | --- | --- | --- | --- | --- | --- | --- |
| | Snow Ice | Clear Ice | Snow Ice | Clear Ice | Snow Ice | Clear Ice | Flooded Snow |
| **Site 002** | 0.01 | 0.27 | 0.08 | 0.19 | 0.03 | 0.40 | |
| **Site 003** | 0.05 | 0.28 | 0.01 | 0.32 | 0.02 | 0.36 | X |
| **Site 004** | 0.01 | 0.28 | 0.00 | 0.51 | 0.10 | 0.32 | X |
| **Site 005** | 0.06 | 0.32 | 0.17 | 0.16 | 0.18 | 0.36 | |
| **Site 006** | 0.00 | 0.29 | 0.00 | 0.32 | 0.04 | 0.37 | X |
| **Site 007** | 0.00 | 0.32 | 0.01 | 0.35 | 0.02 | 0.39 | |
| **Site 008** | 0.09 | 0.25 | 0.06 | 0.33 | 0.03 | 0.40 | X |

### 2.5.1 Dry Snow Conditions


The first experiment reflected dry snow conditions that were observed on Lake Oulujärvi on January 29 – 30 and is represented by IOPI in **Figure 3**. The constant values determined from the field data can be found in **Table 2**. Snow and clear ice thickness varied depending on the site SMRT was run for. The site-dependent values can be found in **Table 3**. Snow density values were determined by averaging the snow bulk density measurements. RMSH and correlation length of the air-snow interface were

not varied for these conditions as the impact of this interface was assumed to be negligible due to the relatively small difference in permittivity between air and dry snow. The RMSH for the air-snow interface was set to 1 mm and the correlation length to 10 mm. The correlation length of the snow-ice interface was set to 50 mm.





While the above parameters were available for dry snow conditions, information on the RMSH and interface correlation length

for both the snow-ice interface and ice-water interface, snow ice porosity, and snow ice bubble radius were unknown. Therefore, these properties were varied to determine the optimal values for the ice cover on Lake Oulujärvi. The ranges of these properties can be found in **Table 4**. These ranges were selected due to past observations of snow-ice RMSH (Wakabayashi et al., 1999) and past experiments conducted using SMRT that showed likely ranges for deeper lakes (Murfitt et al., 2022).

**Table 4.** Tested ranges for unknown properties under dry snow conditions.

| Unknown Property | Property Range (Interval) |
|---|---|
| Snow-Ice RMSH (mm) | 1.0, 2.0, 3.0 |
| Ice-Water RMSH (mm) | 0.5 – 2.5 (0.01) |
| Ice-Water Correlation Length (mm) | 10, 30, 50, 70, 100 |
| Snow Ice Bubble Radius (mm) | 0.5, 1.0, 1.5, 2.0 |
| Snow Ice Porosity (%) | 1, 3, 5, 7, 10 |

### 2.5.2 Wet Snow Conditions: Varying Depths

The second experiment reflects wet snow conditions that were observed on Lake Oulujärvi on March 1 and 2, and are represented by IOPII, IOPIIa, and IOPIIb in **Figure 3**. There were three different conditions tested in this experiment. The first

assumed the entire snowpack had the same VWC (representative of IOPII). The second assumed that only the top layer of the snowpack contained water (IOPIIa). The final condition assumed that the layer of snow directly on the ice cover contained water (IOPIIb). This reflects observations from the field which indicated that on March 1 the top of the snow was wet but by March 2 the water had percolated through the snowpack and was present in the lower layers. These observations are reflected in the changes in layer temperature recorded. For IOPII, snow pit density values were averaged across both days. SSA values

were also determined by averaging all observed values. For IOPIIa, density and SSA measurements were determined by averaging layers where observed snow temperatures were >=273.15 K. The temperature, density, and SSA for the bottom layer of the generalized snowpack were determined using layers where the temperature was <273.15 K. This was done to reflect observations of a wet top snow layer during IOPIIa and matches snow pit observations where the temperatures of the upper layers were consistently above melting. For IOPIIb, density and SSA of the lower layer were averaged for layers where

observed snow temperature was >=273.15 K. The properties for the upper layer were determined by averaging temperatures, density, and SSA for layers with temperatures <273.15 K. This reflected observations on March 2 that indicated the lower layers of the snowpack were wetter and that the temperatures of lower layers were consistently at the melting point. In addition, simulations using different densities were assessed with a range of 225 to 450 kg m$^{-3}$ at an interval of 5 kg m$^{-3}$ to explore the




impact of changing snow densities on backscatter under wet conditions. **Table 2** shows the constant snow and ice properties
for these experiments. **Table 3** shows the ice thickness data for each of the sites. The optimal values for ice properties (interface
correlation length, bubble radius, and porosity) from the dry snow experiment were continued forward.

Root mean square height values near the optimal RMSH were used to explore whether variations would impact backscatter
when VWC of the snowpack changed. The ranges for the varied parameters are shown in **Table 5**. The same ranges were
applied to all three conditions, however, the interface that was focused on differed. The selected values for the roughness
parameters (RMSH and correlation length) of air-snow and snow layer boundaries are derived from previously published
studies of snow roughness over both land and sea ice due to the limited observations of these parameters on lake ice (Landy et
al., 2019; Komarov et al., 2017; Petrich and Eicken, 2010; Dinardo et al., 2018; Baghdad et al., 2000). For these experiments,
Sentinel-1 observations are used to investigate the values of the parameters likely to result in the observed backscatter values.

**Table 5.** Tested ranges for different properties for IOPII, IOPIIa, and IOPIIb simulations.

| Tested Property | Property Range (Interval) |
|---|---|
| **All Snow Layers Wet (IOPII)** | |
| RMSH Air-Snow Interface (mm) | 0 – 5.0 (0.25) |
| Correlation Length Air-Snow Interface (mm) | 10, 30, 50, 150 |
| RMSH Snow-Ice Interface (mm) | 0.5 – 4.0 (0.5) |
| RMSH Ice-Water Interface (mm) | 1.0, 2.0 |
| Volumetric Liquid Water Content (%) | 0.02 – 1.00 (0.02) & 0 – 20.00 (1.00) |
| **Single Snow Layer Wet (IOPIIa & IOPIIb)** | |
| RMSH Wet Snow Layer Boundaries (mm) | 0 – 5.0 (0.25) |
| *Air-Snow for IOPIIa and Snow-Snow for IOPIIb* | |
| Correlation Length Snow Layer Boundaries (mm) | 10, 30, 50, 150 |
| RMSH Snow-Ice Interface (mm) | 0.5 – 4.0 (0.5) |
| RMSH Ice-Water Interface (mm) | 1.0, 2.0 |
| Volumetric Liquid Water Content (%) | 0 – 1.00 (0.02) & 1.00 – 20.00 (1.00) |





### 2.5.3 Wet Snow Conditions: Saturated Layer

The final experiment focuses on the observations made on March 25 – 26 which is represented as IOPIII in **Figure 3**.
Observations on these dates indicate that there was a substantial amount of water present between the snow and the top of the ice. This experiment represents this condition by adding a thin 0.04 m saturated snow layer on top of the ice. The water fraction for the saturated layer is 63% and the remainder is ice, this could also be thought of as a slush layer that has been observed for lakes at mid-latitudes (Ariano and Brown, 2019). Only one high value of water fraction was tested to focus on how changes in other properties impacted backscatter under these conditions. Due to capillarity, the dry snow layer above the saturated layer
was also wet. Therefore, additional tests were conducted using varying VWC, 1%, 2.5%, 5%, for a 0.04 m snow layer above the saturated layer. Following a similar process to the conditions established for the variable wet snow layers (IOPIIa and IOPIIb), snow pit data for layers where the temperature was >=273.15 K were used to determine the density for wet layers of the generalized snowpack. Layers where the temperature was <273.15 K were used for dry snow layers in the generalized snowpack. Unlike the experiments for IOPII, SSA values were not available for IOPIII and therefore values are determined
from **Equation 2,** and $p_{ex}$ was deduced from **Equation 1**. The snow and ice conditions can be found in **Table 2** and ice thickness values for each site in **Table 3**. As with the second experiment, optimal properties determined under dry conditions were held constant.

Properties that were varied include the RMSH of the air-snow, snow-slush, slush-ice, and ice-water interface as well as the
correlation length of the snow-slush interface. The ranges for these values are shown in **Table 6**. As with the second experiment, Sentinel-1 observations are used to investigate the most likely conditions.

**Table 6.** Tested ranges for different properties for IOPIII simulations.

| Tested Property | Property Range (Interval) |
|---|---|
| RMSH Air-Snow Interface (mm) | 0.0 – 5.0 (1.0) |
| RMSH Slush-Ice Interface (mm) | 0.5 – 4.0 (0.5) |
| RMSH Ice-Water Interface (mm) | 1.0 & 2.0 |
| RMSH Snow-Slush Interface (mm) | 0 – 5.0 (0.25) |
| Correlation Length Snow-Slush Interface (mm) | 10, 50, 150 |





# 3. Results

## 3.1 Dry Snow Conditions

Following the procedure in section 2.5.1, optimal values were selected for RMSH, ice-water interface correlation length, porosity, and bubble radius based on the SMRT run with the minimum average difference between simulated and observed backscatter. The optimal values for the different parameters were found to be 1.26 mm for RMSH, an ice-water interface correlation length of 70 mm, a porosity of 10%, and a bubble radius of 1 mm. Differences between observed and modelled backscatter ranged from 0.25 to 2.36 dB (**Figure 4**), with an RMSE of 1.91 dB for HH-pol, 1.55 dB for VV-pol, and a combined RMSE of 1.74 dB. When using a bubble radius of 1.5 and 2 mm, similar combined RMSE values were found, 1.84 dB and 1.64 dB, respectively. However, the modelled backscatter for the minimum tested RMSH exceeded the minimum observed values from the field sites.

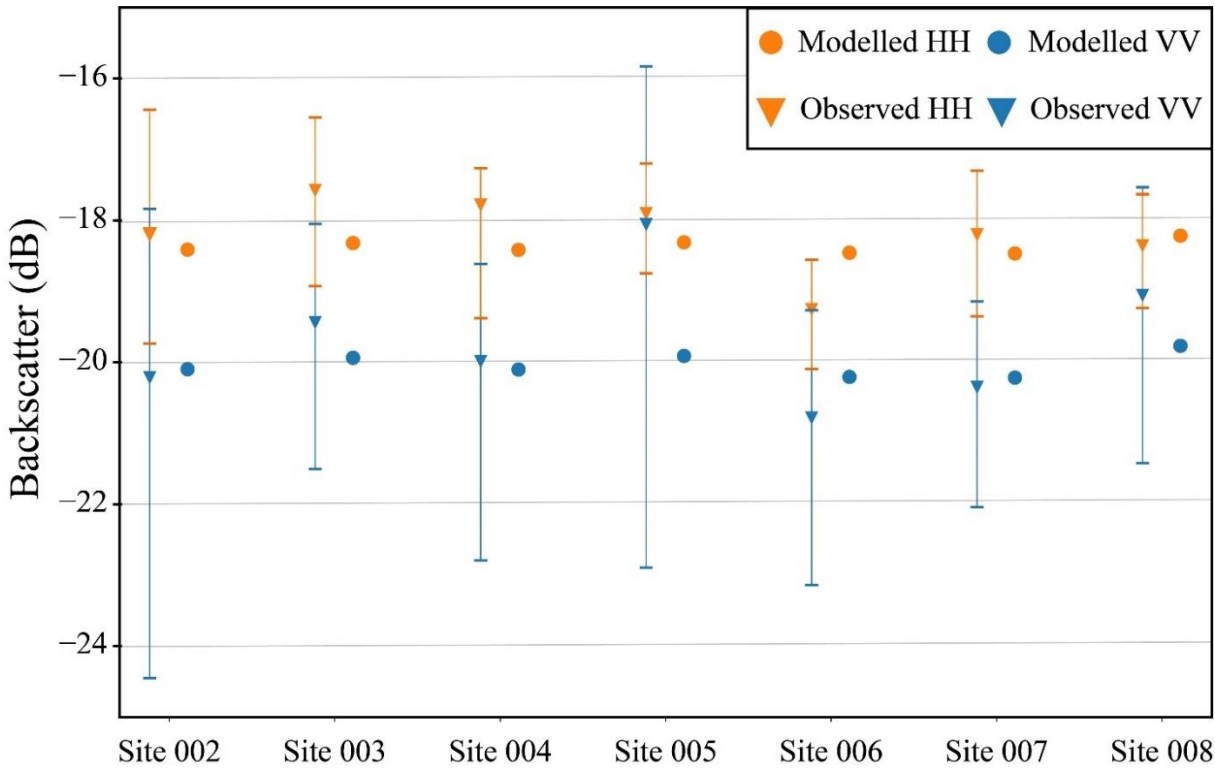

**Figure 4.** Comparison between modelled and observed Sentinel-1 backscatter for IOPI. Vertical lines show the standard deviation in observed backscatter for each site.

**Figure 5** shows the change in modelled backscatter with increasing RMSH at the ice-water interface, for this example sites 006 and 008 are displayed to show a site from IOPI with and without snow ice. These simulations only focused on RMSH as it has been identified as the key property influencing backscatter under dry snow conditions in previous lake ice studies (Gunn





et al., 2018; Murfitt et al., 2022). The red and blue boxes in the figure show the range of Sentinel-1 backscatter values extracted at the different locations. As the RMSH increases to between 1 and 1.75 mm, the modelled backscatter falls within the ranges of observed backscatter for both HH and VV-pol. Additionally, **Figure 5** shows the impact of increasing snow-ice RMSH on

395 backscatter under dry conditions. When an increased value of RMSH is used at the snow-ice interface, the difference between HH backscatter is a maximum of 3.3 dB for site 006 and 2.0 dB for site 008. Modelled HH backscatter is identical above RMSH values of 1.1 mm. The difference between modelled VV backscatter is larger, a maximum of 5.69 dB for site 006 and 4.39 dB for site 008. Similar to HH backscatter, the difference between backscatter values decreases with increasing RMSH and values are identical when RMSH >1.9 mm.

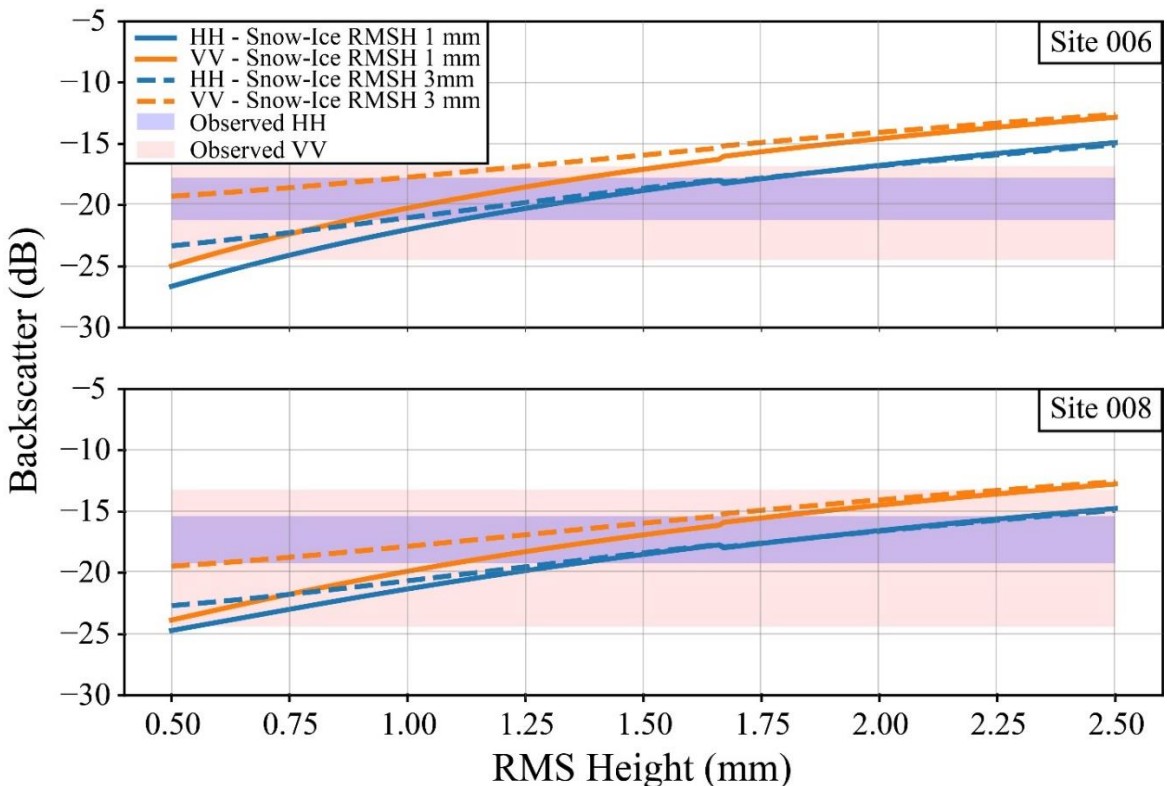

400

**Figure 5.** Modelled backscatter for dry snow conditions (IOPI). For the simulations shown only RMS height of the ice-water interface was varied, all other parameters were held constant. Site 006 shows results where no snow ice was found during the field campaign and site 008 shows results where snow ice was found.

### 3.2 Wet Snow Conditions: Varying Depths

405 **Figure 6** shows the modelled backscatter for the simulation using data from IOPII where all layers of the snowpack have the same VWC. For these simulations VWC was varied as well as the RMSH of the air-snow interface. This was done to explore how changing both the water content and RMSH of the top of the snowpack impacted backscatter. These results are for site 005, the values for other sites are not shown as the resulting backscatter is almost identical when snowpacks are wet. Increasing





VWC by 1% with the top of the snow interface having correlation length of 10 mm results in an average decrease of 9.18 dB
and an average decrease of -11.80 dB for an interface correlation length of 50 mm for all RMSH used (**Figure 6**). This decrease
is largest, >-22 dB, when the surface of the snowpack (air-snow interface) is flat (i.e., has a RMSH of 0 mm). Backscatter is
higher for simulations conducted using an interface correlation length of 10 mm compared to 50 mm. Simulated VV backscatter
is comparable to observed backscatter for conditions with an interface correlation length of 10 mm, RMSH >3 mm, and when
the VWC is above 7.5%. Simulated HH backscatter is lower compared to observed backscatter except when conditions are
modelled with large RMSH and VWC values and when an interface correlation length of 10 mm is used. These experiments
also show that for both polarizations the rate at which backscatter increases slows for VWC >15%. Changes in backscatter in
**Figure 6** are not consistent at a correlation length of 50 mm. This is likely due to IEM, but further investigation is needed.

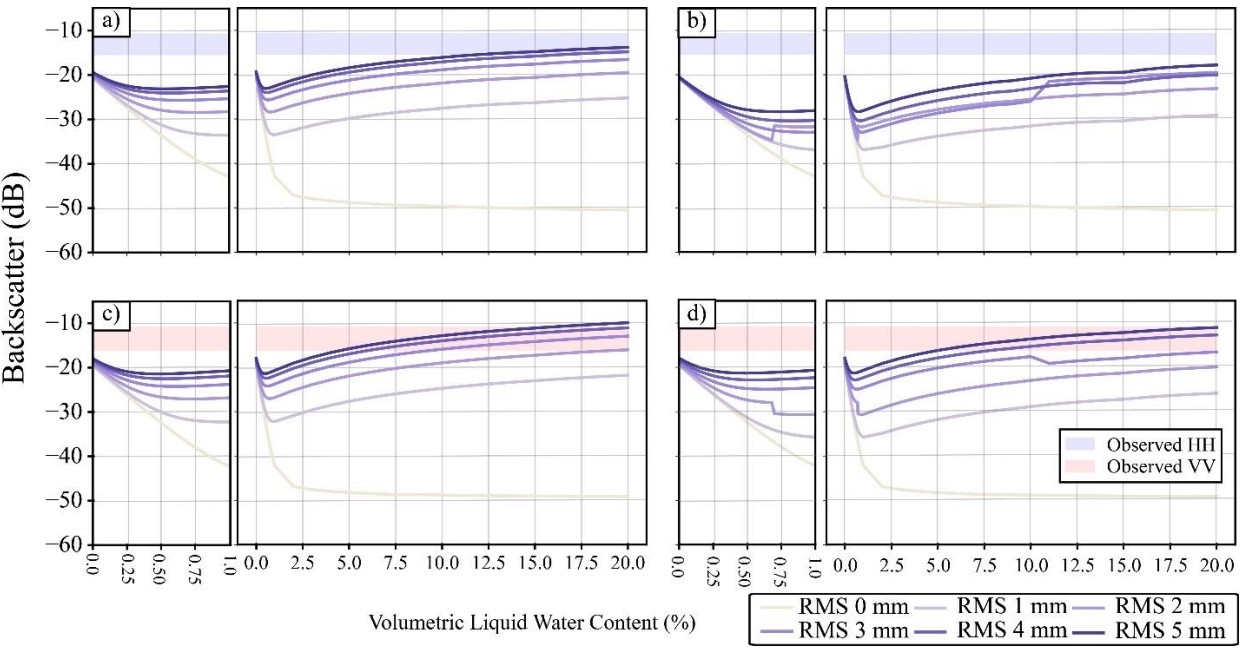

**Figure 6.** Modelled backscatter for site 005 where all layers of the snowpack have the same volumetric liquid water content. Only volumetric
liquid water content and RMSH at the air-snow interface have been varied, all other parameters are held constant. The different lines represent
different values for the RMSH at the top of the snowpack (air-snow interface). a) and c) show results with a correlation length of 10 mm and
b) and d) show results with a correlation length of 50 mm. The blue and red boxes show the observed backscatter from Sentinel-1.

Experiments were also conducted to assess the role that the RMSH of snow-ice and ice-water interfaces have when the entire
snowpack has the same VWC. The RMSH of the air-snow interface was held constant at 5 mm for these simulations. Increasing
either the snow-ice or ice-water interface RMSH has no impact on the backscatter when VWC >0%. There is a slight increase
in backscatter at 0% VWC when a larger snow-ice interface values is used, however, it is small, <1.5 dB. At 0% VWC,
backscatter increases by >3 dB when a larger ice-water RMSH of 2 mm is used. Similar to **Figure 6**, modelled backscatter is
most comparable to observed VV-pol when the interface correlation length is 10 mm and the VWC is >7.5%.



**Figure 7** shows the impact of changing snow density on backscatter when the VWC of the snowpack is equal. This was done to confirm that the primary factors impacting backscatter were RMSH and VWC, not density. For this experiment the ice-water interface RMSH was set as 1 mm and the snow-ice interface RMSH was 2 mm. Similar to **Figure 6**, backscatter increases with higher VWC. Increasing total snow density results in a rate of change of backscatter less than 9 dB over the large range of densities.

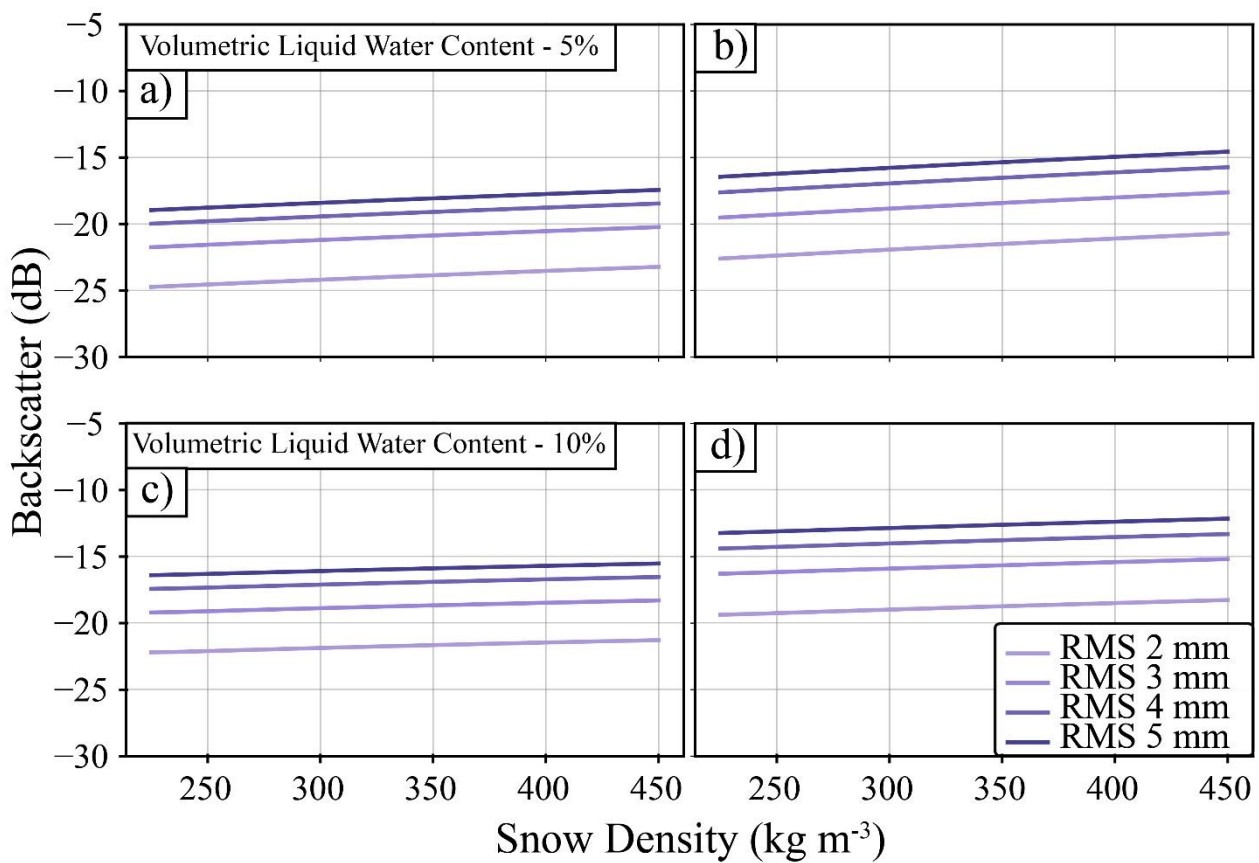


**Figure 7.** Modelled backscatter for site 005 where all layers of the snowpack have the same volumetric liquid water content, the RMSH of the ice-water interface is set to 1 mm and the RMSH of the snow-ice interface is set to 2 mm. The scenario was run for different volumetric liquid water content values and a range of densities. a) and c) show modelled HH results and b) and d) show modelled VV results.

**Figure 8** shows the variations in modelled backscatter when water was added to different layers of the snowpack. These

simulations are identical to **Figure 6**, however, the interface of roughness within the snowpack was changed. For March 1 (IOPIIa), RMSH of the air-snow interface was changed but for March 2 (IOPIIb) the RMSH at the interface between two layers of snow was changed as illustrated in **Figure 3**. For March 1, when the top of the snowpack contained water, only VV backscatter from Sentinel-1 was available, therefore only VV modelled results are shown. Similar to previous experiments, modelled backscatter was closest to observations when VWC >7.5% and an interface correlation length of 10 mm was used.

According to simulations, the RMSH of the air-snow interface would need to be >3 mm to produce backscatter comparable to





the observed values. Overall results are similar to the patterns observed in **Figure 6**, however, there is a smaller decrease in backscatter with the initial addition of water to the snow layer compared to the decrease observed for when all layers of the snowpack contained water. For March 2, where the layer of snow directly on the ice contained water, only HH backscatter from Sentinel-1 was available and only modelled HH backscatter is shown. Modelled HH backscatter is lower compared to

the range of observed HH backscatter. Compared to the change in VV backscatter, when RMSH >0 mm, there is a smaller decrease with increasing VWC. Similar to the initial experiments conducted when water was present throughout the snowpack, modelled backscatter for March 1 and March 2 saturates at high levels of VWC.

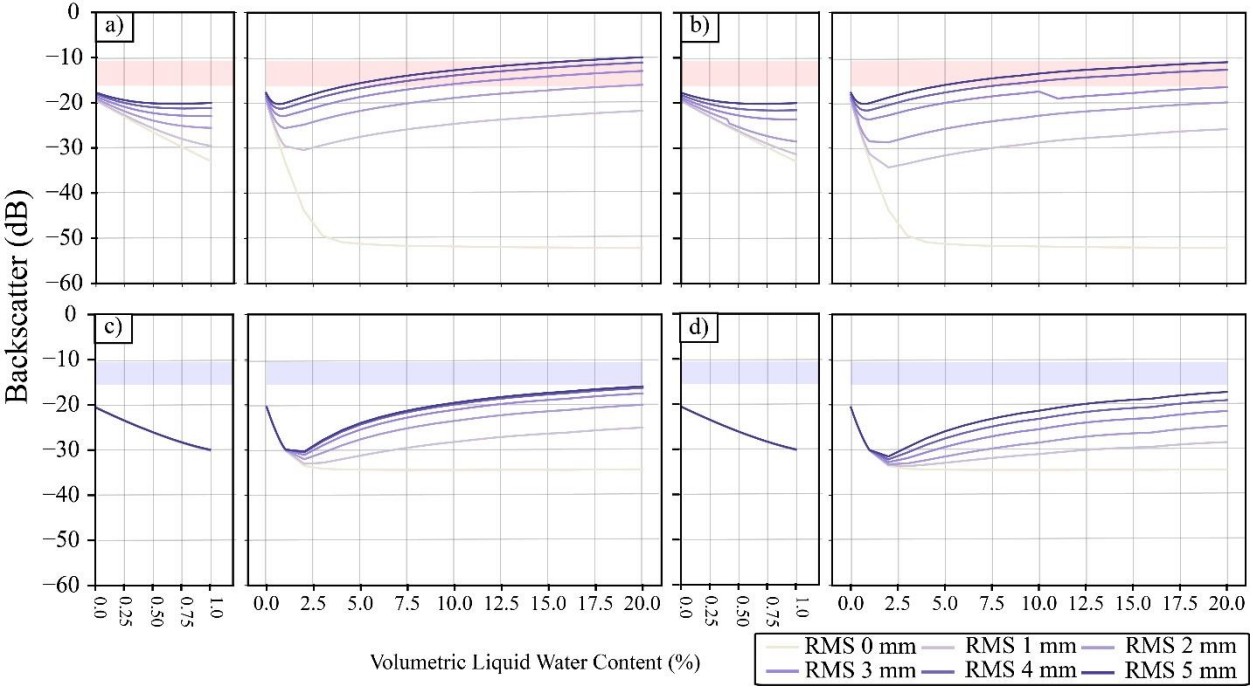

**Figure 8.** Modelled backscatter for site 005 representing IOPIIa, water in the top layer of the snowpack, and IOPIIb, water in the lower layer
of the snowpack. For IOPIIa simulations RMSH at the air-snow interface were changed and for IOPIIb RMSH at the interface between snow layers were changed. The different lines represent different values for the RMSH at the top of the snowpack (air-snow interface). The blue and red boxes show the observed backscatter from Sentinel-1.

### 3.3 Wet Snow Conditions: Saturated Layer

The final experiment added a 0.04 m slush layer at the bottom of the snowpack with fractional water content of 63%. Previous
experiments contained a mix of snow, ice, and water, however, for this experiment the saturated layer only contains ice and water resulting in a higher overall density of 954 kg m$^{-3}$. Additional tests with a wet snow layer over the saturated layers were also conducted. **Figure 9** shows how the modelled backscatter changes with increasing RMSH at the interface between the snow and slush layer for site 006 with snow layers of differing VWC overlying the slush layer. This site was selected as field observations noted that there was a large amount of water located between the snow and the ice surface. Similar to the
experiments in section 3.2, modelled backscatter was identical for the different sites due to the high VWC. The range of



observed backscatter is similar between IOPII and IOPIII, however, modelled backscatter can reproduce observed values at a lower RMSH (<3 mm) for IOPIII compared to IOPII. Additional tests with a wet snow layer were conducted to investigate how the addition of this layer impacted the backscatter from the saturated layer. The range of modelled backscatter decreases as the VWC of the overlying wet snow layer increases. Modelled values when the overlying snow layer has a VWC of >2.5

% fall outside the observed range of both HH and VV backscatter.

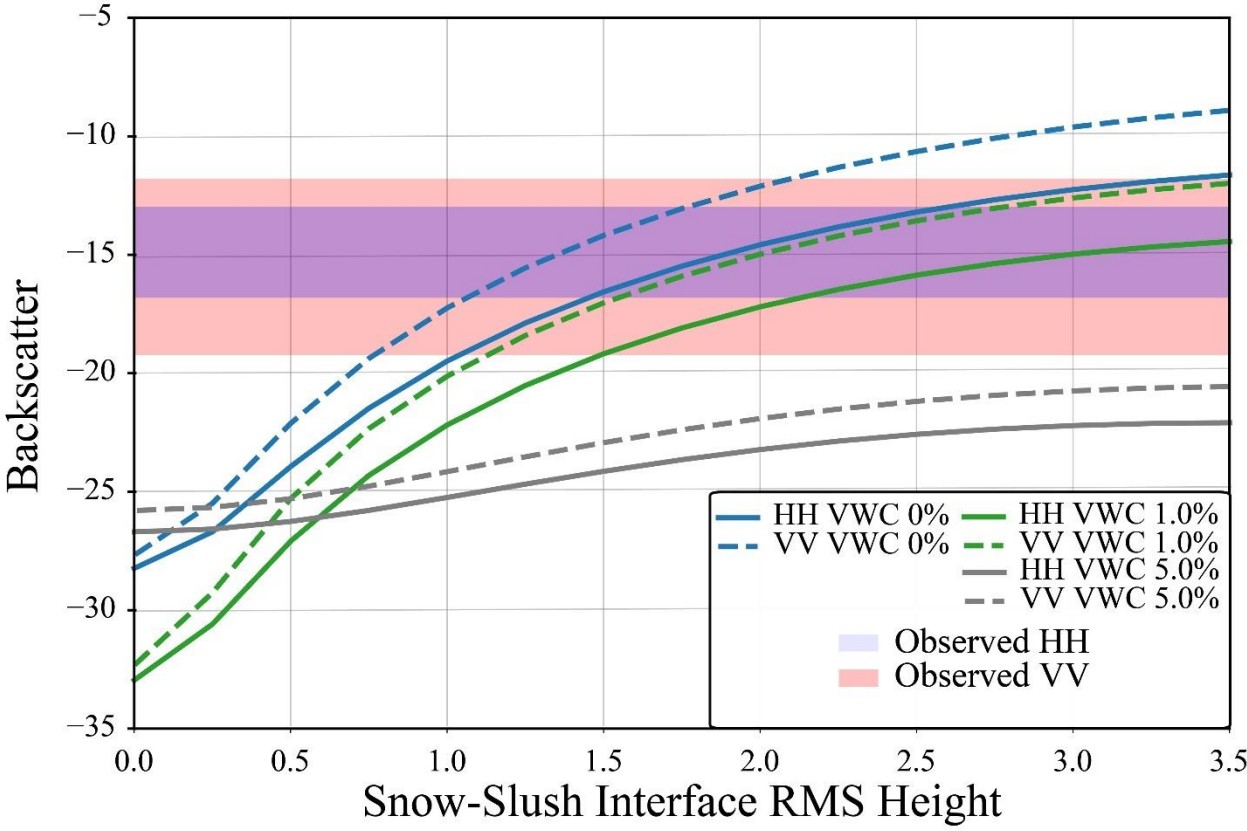

**Figure 9.** Modelled backscatter for site 006 where there is a 0.04 m layer of slush between the lowest snow bottom of the snowpack and the ice surface. All other properties are held constant except the RMSH of the interface between the lowest snow layer and the slush layer. The VWC of the lowest 0.04 m of snow was varied between 0 and 5% The blue and red boxes show the observed backscatter from Sentinel-1.

**4. Discussion**

Recent research has focused on the key role that roughness of the ice-water interface plays in backscatter from lake ice using both observed data and modelling (Atwood et al., 2015; Gunn et al., 2018; Engram et al., 2012). The simulation results from IOPI also indicate that it is likely the main interface for surface scattering is the ice-water interface under dry conditions. However, it should be noted that the results in **Figure 5** show that at lower values of RMSH for the ice-water interface, larger

RMSH at the snow-ice interface causes an increase in backscatter. Yet, field observations indicate that under dry conditions RMSH values of 1 mm at the snow-ice interface are more likely (Wakabayashi et al., 1999; Han and Lee, 2013). The difference



between modelled backscatter values with different snow-ice RMSH values also reduces when smaller interface correlation lengths are used. For example, the difference at site 006 was 5.69 dB at an interface correlation length of 50 mm but drops to 2.84 dB when an interface correlation length of 10 mm is used. This highlights the importance of properly parameterizing all
aspects of roughness for the difference interfaces, which is notoriously known as difficult. For lake ice, while recent studies have been able to extract roughness using ground penetrating radar with a frequency of 800 MHz (Gunn et al., 2021), no measurements have been acquired at cm or mm scale which are relevant for C-band SAR backscatter. Obtaining accurate in situ data for these measurements is an area of continuing study. While values of RMSH ~1 mm are more likely at the snow-ice interface, higher values are not impossible and can be created when there are deformations in the ice surface. For example,
in **Figure 10**, backscatter for site 005 increases faster compared to the other two sites and is higher prior to IOPII. This is due to the site being located near a crack in the ice surface identified from brighter tones in the Sentinel-1 imagery. Cracks and deformations have been noted to result in higher backscatter by up to 10 dB compared to areas where the ice surface is smoother (Morris et al., 1995). These deformations cannot be modelled by SMRT. **Figure 10** also supports recent conclusions that suggest ice-water interface roughness increases rapidly at the start of the ice season but then become stable as ice growth slows
(Murfitt et al., 2023). Backscatter for sites 006 and 008 remains stable between IOPI and the end of February. With recent sensitivity analysis indicating that backscatter change is primarily driven by increasing RMSH (Murfitt et al., 2022), it is likely that RMSH of the ice-water interface varies little during this period for these sites.

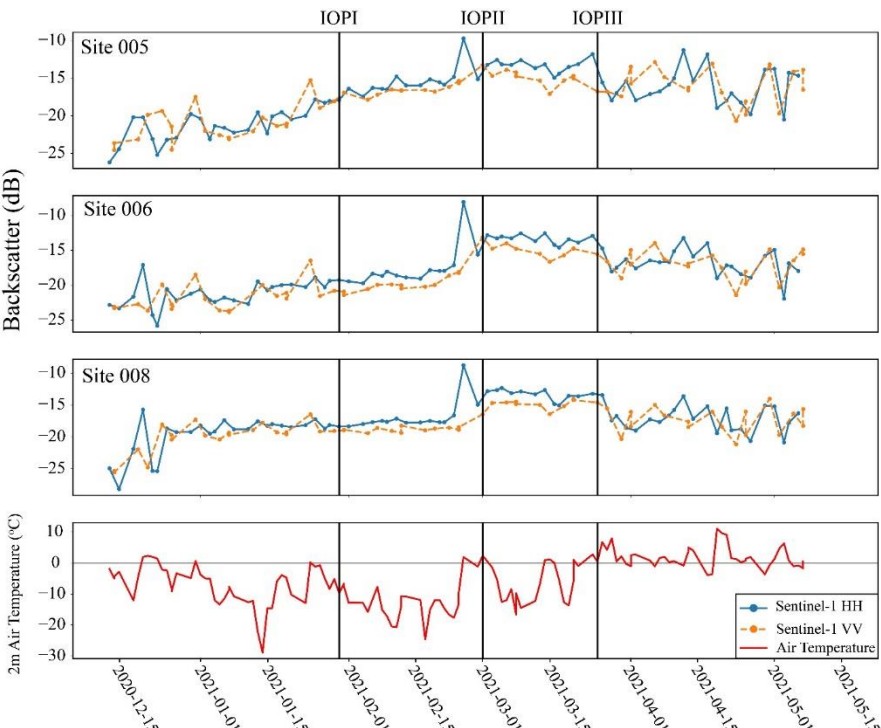

**Figure 10.** Backscatter evolution for selected sites on Oulujärvi and 2-m air temperature from ERA5 reanalysis data.





Under wet conditions, the experiments indicate that it is likely that the dominant interface for surface scattering switches from
        the ice-water interface to the top of the snow layer with the highest VWC. This is demonstrated by experiments which showed
        no difference in modelled backscatter when RMSH of the snow-ice and/or ice-water interface is increased. This agrees with
        modelling work conducted in South Korea which demonstrated that when the ice surface was wet it was the primary scattering
        interface but the contribution decreased as the surface froze (Han and Lee, 2013). **Figure 6** and **Figure 8** both demonstrate
that in simulations the addition of RMSH at the top of the wet snow layer causes a change in the response of backscatter to
        increasing VWC. When the interface is smooth, backscatter decreases with increasing VWC. This is commonly reported in
        lake ice literature, with backscatter decreasing during break-up as snow melts and the ice cover decays (Antonova et al., 2016;
        Duguay et al., 2002; Murfitt and Duguay, 2020). The decreasing backscatter is a result of the higher liquid water content
        increasing the absorption of the microwave signals at these layers. The explanation for the increasing backscatter with the
increase in VWC and RMSH is due to the increasing contrast in permittivity at the surface (or between dry and wet layers).
        The value of the real permittivity component for dry snow is 1.41, similar to that of air meaning there is little reflection
        occurring at the interface between these two surfaces. However, when water is added to the snowpack, the permittivity
        increases to 2.34 when VWC is 5% and 5.46 when VWC is 20%. The increased permittivity results in an increase in the single
        backscattering component from IEM. The increasing roughness at the interface will also increase the simulated diffuse
scattering. This results in the modelled backscatter from the wet and rough surface increasing. While there has been limited
        investigation of these interactions for lake ice, these results are expected based on modelling conducted for terrestrial snow
        which shows increasing backscatter with increasing water content at increasing values of RMSH (Baghdad et al., 2000; Shi et
        al., 1992; Nagler and Rott, 2000; Mätzler and Schanda, 1984). Furthermore, terrestrial snow experiments demonstrate that
        increasing water content leads to surface scattering being dominant at all incident angles, supporting the results observed in
SMRT experiments (Shi et al., 1992).

        Observations support the increase in backscattering because of melt under certain conditions. Between IOPI and IOPII there
        is an increase in observed backscatter (**Figure 10**). The results of the simulations indicate that this is likely due to a combination
        of interface roughness and increasing water content. This change in interface roughness is likely a result of a melt-freeze event.
The occurrence of this event is supported by the temperature profiles shown in **Figure 11**. Prior to data collection for IOPII
        between February 25 and 28, hourly air temperature from ERA5 reanalysis data fluctuated between 2.31 °C and -2.52 °C.
        Temperatures also fluctuated between March 1 and March 2 when IOPII data was collected, with maximums of 3.97 °C on
        March 1 and minimums of -2.64 °C on March 2. The fluctuations in temperature could result in the higher backscatter observed
        on IOPII. Simulations appear to provide support for this in **Figure 6** and **Figure 9** which show that the increased VWC and
RMSH are both necessary to model the observed backscatter for these dates. However, these conditions may not be true at all
        sites as some sites did report lower backscatter on March 1. **Figure 11** shows the Sentinel-1 imagery for the images proceeding
        and including IOPII (February 25, 28, March 1, and 2, 2021). Sites 003, 007, and 008 are all located over areas of decreased
        backscatter with average values ranging from -14.3 to -16.5 dB compared to backscatter ranging from -12.4 to -13.1 dB for


other sites. The darker tones observed for these sites could indicate that the change in RMSH was lower or that less water was

present in the overlying snow layer at this site. Observations of backscatter from other lakes indicate that this increase is not

uncommon for melt events (Antonova et al., 2016; Murfitt et al., 2018, 2023).

**Figure 11.** The change in SAR backscatter for available images between February 25 and March 2, 2021. Data are from Copernicus Sentinel-1 (2021), processed by ESA.






The above explanation supports the increase in backscatter between January/February and March. However, when temperatures become lower, a minimum of -26.60 °C between March 5 – 13 and a minimum of -14.16 °C between March 17 – 20, backscatter remains similar to the values observed during IOPII. This is unexpected as the return to colder conditions indicates that conditions should become drier, and the ice-water interface would become the dominant control over backscatter. However, when backscatter values for two dates, March 14 and 19, are compared to Sentinel-1 observations during IOPI there is an average increase in backscatter of 5.16 dB for HH-pol and 3.69 dB for VV-pol. This suggests that mid-winter melt-freeze events have a lasting impact on the ice conditions and backscatter from lake ice. There are likely explanations for this increase, the first is that the melt-refreeze caused an increase in the snow ice thickness. The refreeze of water from the melt event prior to and during IOPII at the ice surface causes an increase in the thickness of the snow ice layer, this is supported by field measurements. Due to the presence of large spherical scatterers, the snow ice layer has a strong scattering behavior. If larger spherical bubbles were formed this would also contribute to a higher backscatter, however, simulations suggest that the increase in bubble radius causes a weak increase of <2 dB. Another possible explanation is that the melt-freeze event resulted in an increase in the RMSH at the snow-ice interface or at the ice-water interface through the formation of ripples or dunes on the underside of the ice sheet (Ashton, 1986). However, this cannot be confirmed for Lake Oulujärvi. As a conclusion, it is difficult to fully elucidate the cause of the sustained high backscatter during the refreezing after the March melt-event, but it reveals that internal changes in the snow and ice likely occurred during this period. Further research is required to explore how melt-freeze events impact the lake ice properties and the influence this has on backscatter.

The highest backscatter values were obtained when simulations were conducted using a saturated layer between the bottom of the snowpack and the ice column with either dry snow or a layer of snow with a low VWC (<1%) overlying it. These larger values are related to the increased permittivity of the saturated layer, 22.47, compared to the previous observations noted for wet snow layers. According to the backscatter evolution in **Figure 10**, values for IOPIII are lower compared to IOPII. This is likely a result of both increased water content and lower RMSH between the slush layer and the bottom of the snowpack, which as with IOPII, results in increased simulated diffuse scattering at the saturated interface. The Sentinel-1 images in **Figure 12** support this with darker tones around the edge of the ice and throughout the ice cover being an indication of decay. **Figure 10** shows that beyond March 26 there is large fluctuations in backscatter patterns as temperatures are around 0 °C and result in melt-freeze cycles discussed above. The large backscatter values produced through the saturated layer experiments also provide the most likely explanation for the spike in backscatter that occurred on February 25, reaching a maximum of -7.58 dB (**Figure 10 and 11**). The experiments conducted for IOPII were not able to produce these values of backscatter, however, the saturated layer experiments were able to achieve similar values. Similar bright radar returns have been observed for areas of slush in past side-looking airborne radar images from Manitoba, Canada (Leconte and Klassen, 1991). Bright tones were connected to layers of slush under dry layers of ice and snow leading to an increased contrast in the permittivity of the layers and higher returns (Leconte and Klassen, 1991). This is a likely explanation for the spike observed on February 25 and is supported by the results of the modelling for IOPIII. However, it is important to note that the simulations also showed that



when a layer of snow with a higher VWC (>2.5%) is overlying the slush layer, it becomes the dominant surface. Therefore, accurate information on the VWC throughout the snowpack is crucial and further work is needed on studying the complexity of slush events.

580 **Figure 12.** The change in SAR tones for available images between March 21 and March 26, 2021. This figure contains Copernicus Sentinel-1 data (2021), processed by ESA.



The main limitation of this study is the creation of a generalized snowpack, which may differ from reality. The generalized snowpack was used to simplify the modelling set up and address variations across the different sampling sites. However, using a 2-layer approach meant that differences in snow morphology (i.e., grain type) were largely ignored. Additionally, other features in the snowpack, such as highly scattering ice lenses and pipes, were not included. While this limitation could introduce error for the dry snow simulations, as demonstrated by the IOPII experiments, differences in density and volume scattering in general had little impact on the backscatter observed from the wet snowpack (**Figure 7**). Another issue encountered in the modelling was the underestimation of HH backscatter throughout the experiments. This was observed regardless of if modelling was conducted under dry or wet snow experiments. One possible explanation for this underestimation is that VV backscatter is more responsive to changes in the roughness of the surface when represented using IEM (Fung and Chen, 2010). This is supported by the results of the experiments where varying different ice properties result in lower HH backscatter compared to VV. Additionally, further collection of snow microstructure data over lake ice is an area of continued interest to better understand the distinction between snow cover on lake ice and terrestrial snow and represent this in modelling experiments.

## 5. Conclusions

This is the first study to explore the impact of changing volumetric liquid water content on backscatter from lake ice using SMRT. Initial experiments conducted assuming dry snow conditions continue to support previous assertions that the ice-water interface plays the key role in controlling the backscatter from lake ice (Gunn et al., 2018; Engram et al., 2013; Atwood et al., 2015). However, direct observations of the basal roughness are needed to understand how and if these changes occur in nature to quantify the impact. Simulations also indicate that it is likely increasing liquid water content in the overlying snow layers that causes the dominant interface to shift from the ice-water to the surface, resulting in higher backscatter, and a major increase / emergence of the role of the surface roughness. The largest backscatter values were produced when slush layers were present overlying the ice column. The increased dielectric constant of the wetter snow layers combined with the higher roughness is likely what results in these higher backscatter values. These patterns are supported by field observations and congruent Sentinel-1 overpasses. Furthermore, it provides support for past explanations of bright returns observed from lake ice surfaces (Leconte and Klassen, 1991).

Further work is needed to continue to improve understanding on how snow and ice properties change under these conditions. While the field observations used in this study provide valuable information, continued study of snow and ice properties such as SSA, RMSH, and interface correlation length both before and after mid-winter melt events will be crucial in parameterizing radiative transfer models throughout the ice season. Also of interest is the impact of these events on the properties of snow ice layers, particularly with the bubble radius and porosity, which have been noted as factors impacting backscatter in past sensitivity analysis (Tian et al., 2015; Murfitt et al., 2022). Improving the parameterization of wet snow





conditions over lake ice will be important for retrieval of properties such as RMSH. Accurate retrieval of RMSH is also crucial for developing algorithms for ice thickness retrieval. Additionally, effective modelling of backscatter under wet conditions is useful in training lake ice classification models to identify areas of weaker ice types or slush that could pose hazardous to users of lake ice for transportation and recreation during winter months. However, further work is needed as the processes are complex and collection of additional field data and observations on conditions are needed to improve the representation of these events in modelling.

**Appendix A**

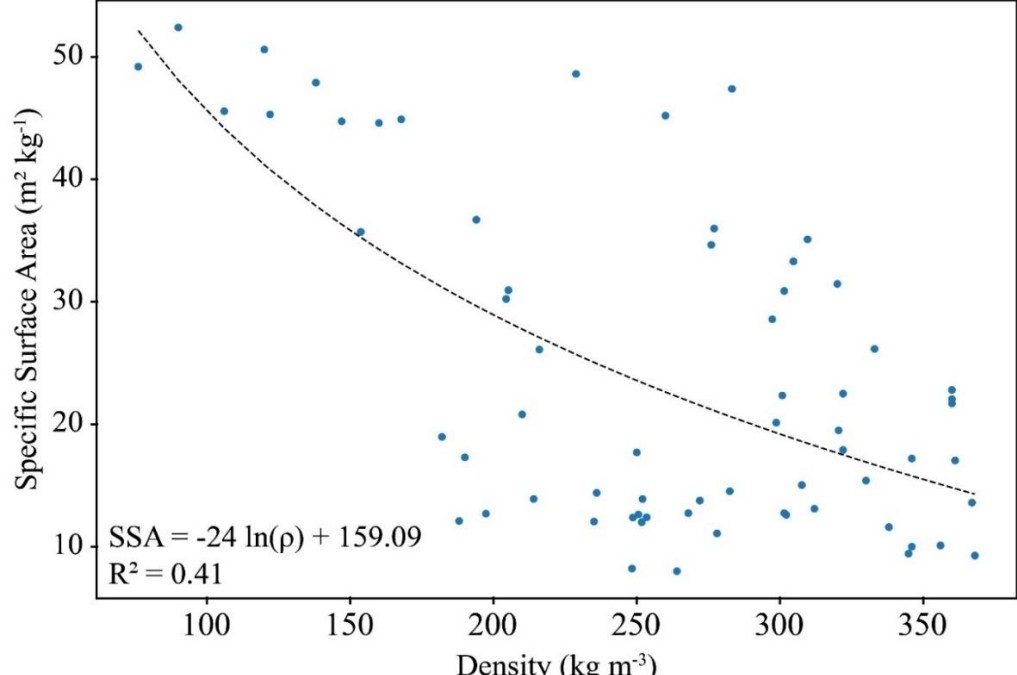

**Figure A1.** Developed relation between density and SSA from samples obtained during all field campaigns conducted on Lake Oulujärvi.

**Code Availability.** The SMRT model code, including the new permittivity formulations, is available from https://github.com/smrt-model/smrt/releases/tag/v1.1.0 (last access: April 11, 2023).

**Competing Interests.** The authors declare that they have no conflict of interest.

**Description of Authors' Responsibilities.** Conceptualization – Justin Murfitt & Claude R. Duguay; Software – Justin Murfitt & Ghislain Picard; Resources and Data Curation – Justin Murfitt & Juha Lemmetyinen; Visualization and Formal Analysis –



Justin Murfitt; First Draft – Justin Murfitt; Revising and Editing – Justin Murfitt, Claude R. Duguay, Ghislain Picard, & Juha Lemmetyinen; Supervision – Claude R. Duguay.


**Acknowledgements.** The authors would like to acknowledge Natural Sciences and Engineering Research Council of Canada, European Space Agency, and EUMETSAT for the financial support provided to this research.

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
