# Peer review of "Forward Modelling of SAR Backscatter during Lake Ice Melt Conditions using the Snow Microwave Radiative Transfer (SMRT) Model"

_The Cryosphere, 2023_

## Author Comment (AC1)

*Thank you for the valuable comments – responses to the comments are included below.*

1. In Figure 2, the time series plot of backscattering reveals higher HH values compared to VV values for IOP I and IOP III. This indicates that volume scattering dominates in these periods. If surface scattering were dominant, higher VV values would be expected, as demonstrated in the modeling work shown in Figure 5. Another approach to verify this is by examining the cross-polarization (cross-pol) of the Sentinel data. In the case of volume scattering, cross-pol values are significantly higher than those of surface scattering. Therefore, during the dry snow period, solely considering the surface effect is insufficient; additional factors such as volume scattering need to be taken into account.

   *Under dry conditions, the contribution of volume scattering is likely very minimal from the dry snow layer. There may be some minor contributions from the lake ice layer. However, past research (e.g., Gunn et al., 2018) has illustrated using polarimetric decomposition that when thin layers of bubbled ice are present the contribution of volume scattering is negligible.*

   *To address the missing cross-pol data and linkages to volume scattering, a short statement was added to the discussion on the patterns observed using Sentinel-1 cross-pol data. Figure 2 was updated to also included these values, "Additionally, while SMRT underestimates cross-pol backscatter, the observed HV and VH backscatter from Sentinel-1 also supports the dominance of the surface scattering regime under dry snow conditions. As shown in Figure 2, between January 1 and February 25, the average HV and VH backscatter was -27.84 and -25.44 dB, respectively. The low cross-pol backscatter values observed indicate that snow ice, or ice likely to depolarize signals, is limited, indicating less volume scattering during these dry conditions (Gunn et al., 2017). This is in agreement with past polarimetric decomposition experiments, which show that volume scattering contributes less compared to surface scattering when these ice types are less present (Gunn et al., 2018)."*

2. On page 4, line 123, it is mentioned that the current SMRT model does not implement cross-polarization calculation. However, despite this limitation, including the measurement Sentinel data of cross-pol would still be valuable in enhancing our understanding of the scattering mechanism. Therefore, I recommend adding cross-pol information to Figure 2. Additionally, to provide a comprehensive reference for backscattering levels, it would be beneficial to include data from the entire season, including the ice-off period, pre-season, and post-season. This would offer a more complete perspective.

   *Figure 2 has been modified to include cross-pol data. Additional images were added to provide a complete picture of the backscatter from before and after the ice season/during melt.*

[Figure]

3.  On page 4, line 128, the paragraph is somewhat confusing. The initial sentences address the dry snow condition, where the snow is considered a two-mixture random medium consisting of ice grains and air. To provide clarity, the sentences need to be rephrased. Subsequently, the paragraph transitions to discussing the wet snow condition. However, it then reintroduces the SHS model and exponential model, which are specifically applied to the dry snow condition. I recommend reorganizing this paragraph to separate the discussion of the dry and wet snow conditions and clearly specify the models chosen for each layer. Additionally, it should be noted that all the assumptions and models described in the original Picard's paper for the wet snow condition pertain to the passive microwave remote sensing regime, which calculates brightness temperature. Applying them directly to backscattering may not yield accurate results.

    *This paragraph was split into two separate paragraphs to clarify the statements, one addressing the electromagnetic model (IBA for dry snow and the addition of MEMLS for wet snow conditions). The other paragraph outlines the microstructure models. We also clarified that sticky hard spheres was used only for ice mediums, while the exponential microstructure model was used for snow mediums. We also added a sentence identifying the limitation of Picard et al. (2022) in the application to the experiments conducted here, "It should be noted that Picard et al. (2022a) specifically evaluate the models for passive microwave analysis, and these models have not been fully verified for active microwave."*

4. On page 9, line 243, it should be noted that the remote sensing data utilized in this paper is backscattering. The temperature of each layer has minimal impact on backscattering and should be clarified from the outset. Consequently, it is not necessary to consider temperature as a factor in the tables and in-situ measurements. Furthermore, I suggest using the term "snow media correlation length" instead of "Pex" in the sentence to maintain consistency and clarity.

*Temperature does have a minimal impact on the dielectrics and the resulting backscatter, but it is needed to generate the most realistic representation of the mediums observed in the field. Two sentences have been added to address this comment, "It is important to note that the temperature of the layers has a minimal impact on the dielectric values and the resulting backscatter but does provide a more realistic representation of the different mediums within the model. Therefore, the temperature data from the field campaigns and CLIMo are included."*

5. Figure 3 illustrates the modeling approach for a multi-layer structure during three IOP periods. To enhance clarity, it would be beneficial to consolidate the information regarding the selected model and input parameters for each layer in a single location. Currently, this information is dispersed across sections 2.1 and 2.5, making it challenging to piece together. It would be helpful to provide a clear explanation of the distinctions between dry snow, snow ice, and pure ice. It seems that all three conditions involve a mixture of ice and snow, differentiated by varying volume fractions. Please provide further elaboration on this matter.

*The same electromagnetic and microstructure models were used for all snow and ice layers; there was no change between wet snow/dry snow or clear ice/snow ice. Additionally, all the necessary input information (layer thickness, temperature, density, etc.) can be found in Tables 2 and 3, Figure 3, and Section 2.5.1 – 2.5.3. An additional table was added which summarizes the microstructure and electromagnetic model information from section 2.1 at the start of section 2.5.*

**Table 2.** SMRT microstructure and electromagnetic model settings used for dry and wet snow layers.

| Physical Layer | Microstructure Model | Electromagnetic Model |
|----------------|---------------------|----------------------|
| Dry Snow | Exponential model | IBA |
| Wet Snow | Exponential model | MEMLS V3 |
| Snow Ice | Sticky Hard Spheres | IBA |
| Clear Ice | Sticky Hard Spheres | IBA |

6. As previously mentioned, the temperature of each layer has a minimal effect on the backscattering calculation compared to the brightness temperature. Therefore, the introduction of the CLIMo model on page 9, line 250 does not appear necessary for this purpose.

*Similar to the comment above, this is to provide the most realistic representation of the medium conditions for the conducted experiment, "It is important to note that while the temperature of the layers has a minimal impact on the dielectric values and the resulting backscatter, it is still needed to run SMRT and provide a realistic representation of the different mediums within the model. Therefore, the temperature data from the field campaigns and CLIMo is essential."*

7. Kindly provide further elaboration on the snow ice porosity. Does a 10% snow ice porosity mean that 10% of the volume consists of air and 90% consists of pure ice? To enhance clarity, I recommend color coding each number to clearly indicate which parameters are derived from in-situ measurements and which ones are based on ad-hoc best fit parameters. This would help differentiate between the two sources of data and improve the overall understanding of the parameter selection process.

   *An additional statement was added to clarify the definition of porosity in relation to snow ice: "Additionally, the porosity of snow ice relates to the ratio of air and ice within the medium; for example, a porosity of 10% indicates that 10% of the medium is air and 90% is ice.". As suggested, n Table 2, values that are not from field measurements were bolded.*

8. Figure 8 requires additional description to improve clarity. Is (a)(c ) referring to IOPIIa, and (b)(d) referring to IOPIIb? It would be helpful to provide clarification regarding the color blocks. Does red represent VV, while blue represents HH? Furthermore,  when comparing Figure 8 (c) and (d): despite varying VWC values from 0 to 1%, the backscattering remains almost identical and shows no sensitivity to RMSH or correlation length. This suggests that surface scattering is not the dominant factor in the overall backscattering for that particular case.

   *Additional labels were added to Figure 8 in order to clarify the components of the figure.*

[Figure]

*While the backscatter does not increase when VWC is on a small scale, it does as VWC increases. A short statement was added addressing this difference: "Additionally when the wet snow interface is within the snowpack (IOPIIb), the RMSH of the interface does not result in a difference in the magnitude of backscatter (Figure 8c and 8d)."*

Minor comments:

1.  Please keep the color consistent through out the figures. Eg. Figure 2/5/10 using blue for HH and orange for VV. But figure 4 use orange for HH and blue for VV

    *Figure 4 has been corrected to match the other figures.*

2.  For figure 5,6,8,9, the color block of the observed HH/VV. Is the max-min range of the HH/VV or the standard

    *The coloured areas represent the minimum/maximum range for the respective polarizations; this has been clarified in the figure captions.*

---

## Author Comment (AC2)

*Thank you for the valuable comments – responses to the comments are included below.*

1. In Figure 2, the time series plot of backscattering reveals higher HH values compared to VV values for IOP I and IOP III. This indicates that volume scattering dominates in these periods. If surface scattering were dominant, higher VV values would be expected, as demonstrated in the modeling work shown in Figure 5. Another approach to verify this is by examining the cross-polarization (cross-pol) of the Sentinel data. In the case of volume scattering, cross-pol values are significantly higher than those of surface scattering. Therefore, during the dry snow period, solely considering the surface effect is insufficient; additional factors such as volume scattering need to be taken into account.

   *Under dry conditions, the contribution of volume scattering is likely very minimal from the dry snow layer. There may be some minor contributions from the lake ice layer. However, past research (e.g., Gunn et al., 2018) has illustrated using polarimetric decomposition that when thin layers of bubbled ice are present the contribution of volume scattering is negligible.*

   *To address the missing cross-pol data and linkages to volume scattering, a short statement was added to the discussion on the patterns observed using Sentinel-1 cross-pol data. Figure 2 was updated to also included these values, "Additionally, while SMRT underestimates cross-pol backscatter, the observed HV and VH backscatter from Sentinel-1 also supports the dominance of the surface scattering regime under dry snow conditions. As shown in Figure 2, between January 1 and February 25, the average HV and VH backscatter was -27.84 and -25.44 dB, respectively. The low cross-pol backscatter values observed indicate that snow ice, or ice likely to depolarize signals, is limited, indicating less volume scattering during these dry conditions (Gunn et al., 2017). This is in agreement with past polarimetric decomposition experiments, which show that volume scattering contributes less compared to surface scattering when these ice types are less present (Gunn et al., 2018)."*

2. On page 4, line 123, it is mentioned that the current SMRT model does not implement cross-polarization calculation. However, despite this limitation, including the measurement Sentinel data of cross-pol would still be valuable in enhancing our understanding of the scattering mechanism. Therefore, I recommend adding cross-pol information to Figure 2. Additionally, to provide a comprehensive reference for backscattering levels, it would be beneficial to include data from the entire season, including the ice-off period, pre-season, and post-season. This would offer a more complete perspective.

   *Figure 2 has been modified to include cross-pol data. Additional images were added to provide a complete picture of the backscatter from before and after the ice season/during melt.*

[Figure]

3. On page 4, line 128, the paragraph is somewhat confusing. The initial sentences address the dry snow condition, where the snow is considered a two-mixture random medium consisting of ice grains and air. To provide clarity, the sentences need to be rephrased. Subsequently, the paragraph transitions to discussing the wet snow condition. However, it then reintroduces the SHS model and exponential model, which are specifically applied to the dry snow condition. I recommend reorganizing this paragraph to separate the discussion of the dry and wet snow conditions and clearly specify the models chosen for each layer. Additionally, it should be noted that all the assumptions and models described in the original Picard's paper for the wet snow condition pertain to the passive microwave remote sensing regime, which calculates brightness temperature. Applying them directly to backscattering may not yield accurate results.

*This paragraph was split into two separate paragraphs to clarify the statements, one addressing the electromagnetic model (IBA for dry snow and the addition of MEMLS for wet snow conditions). The other paragraph outlines the microstructure models. We also clarified that sticky hard spheres was used only for ice mediums, while the exponential microstructure model was used for snow mediums. We also added a sentence identifying the limitation of Picard et al. (2022) in the application to the experiments conducted here, "It should be noted that Picard et al. (2022a) specifically evaluate the models for passive microwave analysis, and these models have not been fully verified for active microwave."*

4. On page 9, line 243, it should be noted that the remote sensing data utilized in this paper is backscattering. The temperature of each layer has minimal impact on backscattering and should be clarified from the outset. Consequently, it is not necessary to consider temperature as a factor in the tables and in-situ measurements. Furthermore, I suggest using the term "snow media correlation length" instead of "Pex" in the sentence to maintain consistency and clarity.

   *Temperature does have a minimal impact on the dielectrics and the resulting backscatter, but it is needed to generate the most realistic representation of the mediums observed in the field. Two sentences have been added to address this comment, "It is important to note that the temperature of the layers has a minimal impact on the dielectric values and the resulting backscatter but does provide a more realistic representation of the different mediums within the model. Therefore, the temperature data from the field campaigns and CLIMo are included."*

5. Figure 3 illustrates the modeling approach for a multi-layer structure during three IOP periods. To enhance clarity, it would be beneficial to consolidate the information regarding the selected model and input parameters for each layer in a single location. Currently, this information is dispersed across sections 2.1 and 2.5, making it challenging to piece together. It would be helpful to provide a clear explanation of the distinctions between dry snow, snow ice, and pure ice. It seems that all three conditions involve a mixture of ice and snow, differentiated by varying volume fractions. Please provide further elaboration on this matter.

   *The same electromagnetic and microstructure models were used for all snow and ice layers; there was no change between wet snow/dry snow or clear ice/snow ice. Additionally, all the necessary input information (layer thickness, temperature, density, etc.) can be found in Tables 2 and 3, Figure 3, and Section 2.5.1 – 2.5.3. An additional table was added which summarizes the microstructure and electromagnetic model information from section 2.1 at the start of section 2.5.*

**Table 2.** SMRT microstructure and electromagnetic model settings used for dry and wet snow layers.

| Physical Layer | Microstructure Model | Electromagnetic Model |
| --- | --- | --- |
| Dry Snow | Exponential model | IBA |
| Wet Snow | Exponential model | MEMLS V3 |
| Snow Ice | Sticky Hard Spheres | IBA |
| Clear Ice | Sticky Hard Spheres | IBA |

6. As previously mentioned, the temperature of each layer has a minimal effect on the backscattering calculation compared to the brightness temperature. Therefore, the introduction of the CLIMo model on page 9, line 250 does not appear necessary for this purpose.

*Similar to the comment above, this is to provide the most realistic representation of the medium conditions for the conducted experiment, "It is important to note that while the temperature of the layers has a minimal impact on the dielectric values and the resulting backscatter, it is still needed to run SMRT and provide a realistic representation of the different mediums within the model. Therefore, the temperature data from the field campaigns and CLIMo is essential."*

7. Kindly provide further elaboration on the snow ice porosity. Does a 10% snow ice porosity mean that 10% of the volume consists of air and 90% consists of pure ice? To enhance clarity, I recommend color coding each number to clearly indicate which parameters are derived from in-situ measurements and which ones are based on ad-hoc best fit parameters. This would help differentiate between the two sources of data and improve the overall understanding of the parameter selection process.

   *An additional statement was added to clarify the definition of porosity in relation to snow ice: "Additionally, the porosity of snow ice relates to the ratio of air and ice within the medium; for example, a porosity of 10% indicates that 10% of the medium is air and 90% is ice.". As suggested, n Table 2, values that are not from field measurements were bolded.*

8. Figure 8 requires additional description to improve clarity. Is (a)(c ) referring to IOPIIa, and (b)(d) referring to IOPIIb? It would be helpful to provide clarification regarding the color blocks. Does red represent VV, while blue represents HH? Furthermore, when comparing Figure 8 (c) and (d): despite varying VWC values from 0 to 1%, the backscattering remains almost identical and shows no sensitivity to RMSH or correlation length. This suggests that surface scattering is not the dominant factor in the overall backscattering for that particular case.

   *Additional labels were added to Figure 8 in order to clarify the components of the figure.*

[Figure]

*While the backscatter does not increase when VWC is on a small scale, it does as VWC increases. A short statement was added addressing this difference: "Additionally when the wet snow interface is within the snowpack (IOPIIb), the RMSH of the interface does not result in a difference in the magnitude of backscatter (Figure 8c and 8d)."*

Minor comments:

1. Please keep the color consistent through out the figures. Eg. Figure 2/5/10 using blue for HH and orange for VV. But figure 4 use orange for HH and blue for VV

   *Figure 4 has been corrected to match the other figures.*

2. For figure 5,6,8,9, the color block of the observed HH/VV. Is the max-min range of the HH/VV or the standard

   *The coloured areas represent the minimum/maximum range for the respective polarizations; this has been clarified in the figure captions.*

*Thank you for the valuable comments – responses to the comments are included below.*

Because this work is building off of several recent studies, it feels like some of the details and background are missing which makes it difficult for someone to step into without having read all of the previous literature.  For example, there is quite a bit in the introduction on recent work showing the dominant scattering mechanism is due to the ice-water surface scattering. That could be reduced to a sentence or two with the relevant references. A higher-level summary of radar remote sensing of lake ice – i.e. all of the potential scattering mechanisms and their contribution, which frequencies have worked best, what is the state of the art in terms of detection of lake ice thickness and other properties – would be useful. Other specific examples are given below.

*The introduction/background has been reworked to address this by adding additional detail. The description of recent scattering mechanism work was three sentences long in the original version, and these have remained (L74-L79). Descriptions of all mechanisms were given from L66 to L74 and have been expanded to include further details about volume scattering. Discussion of radar applications to lake ice studies has also been expanded.*

*"The most common frequency for SAR remote sensing of lake ice is C-band, partially due to the availability of sensors that provide C-band imagery as well as the penetration depth, which is less impacted by upper ice layers and snow cover (Gunn et al., 2017). Observations of lake ice using L-band and X-band can provide additional information to C-band; for example, L-band has shown success in monitoring methane ebullition bubbles for lakes in Alaska (Engram et al., 2012). SAR is the most widely used radar remote sensing technology for lake ice studies; however, other technologies such as radar altimetry are being increasingly used to retrieve properties such as ice thickness. This was demonstrated in a recent investigation which used Jason-2/Jason-3 Ku-band waveforms to estimate lake ice thickness for Great Slave Lake in Canada (Mangilli et al., 2022).*

 *In recent years there has been a shift in understanding how active microwave signals interact with lake ice. Scattering mechanisms (double-bounce, volume, and single-bounce/surface scattering) from lake ice cover is a key topic within the lake ice and radar remote sensing literature. Initial investigations of lake ice in the 1980s using X and L-band side-looking airborne radar systems connected high radar returns to the presence of tubular bubbles in the ice, stating that bright signals in the imagery were due to a double-bounce scattering mechanism (Weeks et al., 1981). This double-bounce was created as the radar signal interacted with the vertical tubular bubbles and then with the ice-water interface, where there is a high dielectric contrast between the ice and water. Further investigations using spaceborne C-band systems (ERS-1 and RADARSAT-1) continued to support this theory and quantified the backscatter observed from lake ice (Jeffries et al., 1994; Duguay et al., 2002). In addition, past research also acknowledged the role of bubbles in contributing to volume scattering of radar signals in lake ice (Gunn et al., 2017; Matsuoka et al., 1999). However, these contributions were found to be smaller compared to the double-bounce mechanism. In more recent years, with the advent of fully polarimetric SAR data, new research has analyzed the scattering contributions from lake ice and determined that the dominant mechanism is a single bounce or surface scattering mechanism (Atwood et al., 2015; Engram et al., 2012; Gunn et al., 2018). This is attributed to*

*roughness at the ice-water interface. Explanations for the roughness at this interface include the presence of tubular bubbles in the lower layers of the ice, methane ebullition bubbles, and differing ice growth rates (Gunn et al., 2018; Engram et al., 2012, 2013)."*

There are so many parameters affecting the signal that are being tested in this sensitivity study that it is at times difficult to follow. This is with in situ measurements providing a lot of model input. In the discussion it says it describes the "importance of properly parameterizing all aspects of roughness for the different interfaces" (line 485) and that "accurate information on the VWC throughout the snowpack is crucial" (line 576), but it's not clear what the relative impact of these properties (and others) have on the results. How would these parameters be constrained in a larger scale application of spaceborne SAR data for lake ice?

*While other properties do impact backscatter and are included, this study and others before it has demonstrated that these properties have limited impact on backscatter from lake ice. To address this, a short statement has been added at the start of the discussion to mention the impact and provide an outline for the discussion: "The experiments outlined in this study looked at several snow and lake ice parameters (e.g., roughness, bubble size, snow stratigraphy, ice stratigraphy, snow microstructure properties, and volumetric water content). Several of these parameters are from field data collected during the 2020-2021 ice season, and others such as volumetric water content, RMSH, correlation length, bubble size, and porosity, must be estimated based on past observations. However, from the results of these experiments, it can be seen that the key properties impacting backscatter from lake ice are primarily RMSH and volumetric water content. Other properties have little impact on the backscatter, which is supported by other sensitivity studies (Gunn et al., 2015; Gherboudj et al., 2010; Murfitt et al., 2022). Therefore, the remainder of the discussion focuses on the impact of RMSH and VMC on backscatter and how observed backscatter from Sentinel-1 supports the modelling observations."*

Comments:

Lines 75, 79, 495: There's no Murfitt 2023 citation in the reference list.

*This is strange – in the submitted version, the citation appeared on line 746 "Murfitt, J., Duguay, C., Picard, G., and Gunn, G.: Forward modelling of synthetic aperture radar backscatter from lake ice over Canadian Subarctic Lakes, Remote Sens. Environ., 286, 1–18, https://doi.org/10.1016/j.rse.2022.113424, 2023." This will be verified upon next submission.*

Line 92-93: Sentence starting with "However, these experiments…" is vague. Can you explain in a little more detail what the limitations in snow cover representation were?

*Further detail has been provided, outlining the snow representation in both experiments, "for example, representing snow cover as only one layer (Wakabayashi et al., 1999) or only considering a bare ice surface (Han and Lee, 2013)".*

Figure 1: What is the green dot on the inset map?

*The green dot has been clarified as the Kajaani Airport meteorological station.*

Lines 147-148: RMSH seems to be a key parameter, used extensively to throughout the analysis and results, but the description here is fairly minimal. Later (line 391) RMSH is described as a key property influencing backscatter as demonstrated by previous studies. It would be good to provide more of that background up front.

*Additional background on both IEM parameters has been provided in the SMRT modelling section, "IEM is parameterized using root mean square height (RMSH), which quantifies the amplitude of the vertical variation of the surface (Ulaby and Long, 2014). Root mean square height is also termed the height standard deviation or the differences between random height deviations and the mean height of the surface (Ulaby and Long, 2014). Vertical roughness has been identified as a key parameter for backscatter from lake ice both at the ice surface and ice bottom (Atwood et al., 2015; Han and Lee, 2013). Increases in the RMSH are linked to higher backscatter values at oblique incidence and increased surface scattering. Interface correlation length, which quantifies the horizontal correlation between two points on the rough surface (Ulaby and Long, 2014). It measures how smoothly surface elevation is changing horizontally. While correlation length has also been evaluated, past modelling of backscatter from lake ice indicates that, upon retrieval, correlation length is more consistent than RMSH values (Han and Lee, 2013)."*

Line 176: I assume the 82 EW and 69 IW SAR images make up the 151 Sentinel-1 images acquired, but right now it reads like a list. I would suggest editing it to read "151 Sentinel-1 (C-band, 5.405 GHz), comprised of 82 Extra Wide (EW) swath HH-pol and 69 Interferometric Wide (IW) swath VV-pol, SAR images…"

*This suggestion has been incorporated into the text.*

Line 221: Maybe I missed this, but how was water content measured?

*This was not clear in the original text. Exact water content measurements were not available, and instead, conditions were developed based on first-hand information provided by the team that collected the field data. The text has been modified to reflect this, "Unfortunately, exact measurements regarding water content were not available. Therefore, qualitative observations of snow moisture conditions were used to generate the SMRT experiments outlined in Section 2.5."*

Line 409: Why is one of the values negative (-11.8) and the other not (9.19)? Since you say "decrease" maybe the negative sign isn't needed?

*Yes, the negative sign is not needed. This has been removed from the text.*

Conclusion: There is not a clear take-home message. I recommend that conclusions and relevant findings from this work be clearly stated in this section, which I think would help strengthen the paper.

*The first paragraph of the conclusion focuses on summarizing the results, and a pair of sentences have been added to re-iterate the key message from the paper, "The results of these experiments show how radiative transfer modelling is valuable in understanding the response of backscatter to lake ice conditions under both dry and wet conditions. These experiments highlight the impact*

*that surface roughness has on backscatter and the change in dominant interface with increasing snow water content."*